# Proprioceptive accuracy in Immersive Virtual Reality: A developmental perspective

Irene Valori[1], Phoebe E. McKenna-Plumley[1], Rena Bayramova[2], Claudio Zandonella Callegher[1], Gianmarco Altoè[1], Teresa Farroni[1]*

1 Department of Developmental Psychology and Socialization, University of Padova, Padova, Italy,
2 Department of General Psychology, University of Padova, Padova, Italy

* teresa.farroni@unipd.it

**Data Availability Statement:** All data files are available from the OSF public repository at the following URL (https://osf.io/b3qd4/).

## Abstract

Proprioceptive development relies on a variety of sensory inputs, among which vision is hugely dominant. Focusing on the developmental trajectory underpinning the integration of vision and proprioception, the present research explores how this integration is involved in interactions with Immersive Virtual Reality (IVR) by examining how proprioceptive accuracy is affected by *Age*, *Perception*, and *Environment*. Individuals from 4 to 43 years old completed a self-turning task which asked them to manually return to a previous location with different sensory modalities available in both IVR and reality. Results were interpreted from an exploratory perspective using Bayesian model comparison analysis, which allows the phenomena to be described using probabilistic statements rather than simplified reject/not-reject decisions. The most plausible model showed that 4–8-year-old children can generally be expected to make more proprioceptive errors than older children and adults. Across age groups, proprioceptive accuracy is higher when vision is available, and is disrupted in the visual environment provided by the IVR headset. We can conclude that proprioceptive accuracy mostly develops during the first eight years of life and that it relies largely on vision. Moreover, our findings indicate that this proprioceptive accuracy can be disrupted by the use of an IVR headset.

## Introduction

From the earliest stages of life, we develop physically, psychologically, and socially through the interaction between our genes and the environment. We experience this environment via sensory information which comes from both the external world (*exteroception*) and the self (*interoception*). Exteroception describes sensory information which comes from the environment around us (e.g. sight, hearing, touch), while interoception is the perception of our body and includes "temperature, pain, itch, tickle, sensual touch, muscular and visceral sensations, vasomotor flush, hunger, thirst" and other sensations (p. 655 [1]). This information, which comes from different, complementary sensory modalities, has to be integrated so that we can interact with and learn from the environment. The multisensory integration that follows takes time to develop and emerges in a heterochronous pattern: we rely on the various sensory modalities to

**Funding:** The author(s) received no specific funding for this work.

**Competing interests:** The authors have declared that no competing interests exist.

different degrees at different points in the human developmental trajectory, during which the sensory modalities interact in different ways [2].

## Proprioception: An emergent perception arising from a multisensory process

Both exteroception and interoception drive our discovery of the external world and the self. One important physical dimension of the concept of self is *proprioception*, which has a definition that is particularly complex and debated in the extant literature. Proprioception belongs to the somatosensory system [3] and has traditionally been defined as the "awareness of the spatial and mechanical status of the musculoskeletal framework" which includes the senses of position, movement, and balance (p. 667 [4]). From this perspective, proprioception is the awareness of the position and movement of our body in space and results from the processing of information from muscle, joint, tendon, and skin receptors. It arises from static (position) and dynamic (movement) information, and is crucial to the production of coordinated movements [5]. In general, researchers are now bypassing the study of unimodal sensory processing to focus on multisensory integration processes. While humans rely on somatosensory information to achieve proprioception in blind conditions, vision can lead to proprioception when proprioceptively informative cues are provided. Indeed, specific visual cues can be considered to be proprioceptively informative to the extent that they aid proprioception. For example, research concerning mirror therapy for phantom limb pain indicates that visual representations of the body (e.g. the lost limb) can be manipulated to induce proprioceptive sensations and perception of movement, touch, and body ownership, even with a complete absence of somatosensory input [6]. Moreover, self-motion studies show that global visual landmarks such as the corners of a room appear to be useful for proprioception, while local visual cues such as surrounding objects [7] or homogeneous visual textures and patterns [8] are not.

We now know that proprioception is a complex body consciousness which flexibly emerges from different interdependent sensory inputs, modalities, and receptors. Proprioceptive information is combined with information from the vestibular system, which detects movement of the head in space, and the visual system to give us a sense of motion and allow us to make estimates about our movements [9]. As such, it plays a vital role in everyday tasks such as self-motion.

As regards the development of proprioception, children up to two years of age tend to make significant proprioceptive errors [10]. While several studies have shown that proprioceptive competence is stably developed by eight years of age [11, 12], others support the finding of a longer developmental trajectory for proprioception, observing that 8- to 10-year-old children are less accurate than 16- to 18-year-old adolescents when making proprioceptively guided movements [13]. Moreover, some studies find improvements in proprioceptive accuracy continuing up to 24 years of age [14].

This proprioceptive development seems to be strictly dependent on visual calibration. In general, sensory organization is qualitatively different across development and across different tasks. In infancy and early childhood, vision appears dominant over somatosensory and vestibular information [15]. Between five and seven years of age, visual influence on proprioception shows non-linear developmental differences [16], although this has not yet been widely studied in a broader range of ages [15]. The developmental trajectory of proprioception may be affected by the fact that across childhood, the sections of the body change in terms of size, shape, and relative location. Indeed, the early importance of vision over somatosensory information could be a result of the lack of reliability of somatosensory input, which is highly unstable during these childhood physical changes [2].

## IVR as a method of studying proprioception

Immersive Virtual Reality (IVR) can be used to manipulate vision while the user performs proprioceptive tasks. Through IVR, we can manipulate individual sources of sensory information, be they visual, vestibular, or proprioceptive, which are physiologically bound together. This makes it possible to study the contribution of these individual sensory inputs and of multisensory integration to self-perception and motor control [17].

In IVR, "the simultaneous experience of both virtual environment and real environment often leads to new or confounded perceptual experiences" (p.71 [18]). For example, users can see themselves standing in the empty space between two mountains but, instead of falling, perceive the floor under their feet. Even with a virtual body representation (e.g. visual perception of an avatar) or without the possibility to see one's own body, IVR can alter a user's body schema [19]. In IVR, users are found to decrease their speed and take smaller steps [20] and experience greater difficulties orienting themselves [21]. To orient and move in space in different environments and tasks, people can switch between reference frames related to the body (e.g. proprioception) or to the external world (e.g. vision). It has been suggested that IVR provides unexpected incongruent stimuli and induces a sensory conflict between vision and proprioception which differently affects users (e.g. sometimes causing motion sickness) depending on their dominant reliance on one of these two reference frames [22]. The possibility to make active movements during the interaction with IVR improves proprioception, even without proprioceptively informative visual landmarks [23, 24]. However, despite the importance of the body senses, the physical feedback (derived, for example, from actively walking during the virtual immersion) is not sufficient to eliminate errors in self-motion and spatial orientation while wearing an HMD [25]. These findings show that HMD-delivered IVR has particular visuo-proprioceptive features that can disrupt proprioception in adults.

However, there is a lack of research regarding how IVR features interact with age-related proprioceptive accuracy, visuo-proprioceptive integration, and self-motion. A recent experimental study with children (8–12 years old) and adolescents (15–18 years old) provides evidence about children's use of vision during self-motion in IVR [26]. The authors intentionally created a mismatch between visual feedback (visual flow) and proprioceptive feedback (active motion) during different motor tasks. They measured children's ability to *recalibrate* (to adapt their motor actions to the provided abnormal visual input) and *re-adapt* to the normal characteristics of the real environment. As with adults in previous studies [27, 28], children and adolescents showed the ability to recalibrate in a few minutes. However, children re-adapted to reality significantly more slowly than adolescents, demonstrating more pronounced post-exposure effects. These findings indicate that the motor performance of children, more so than adolescents, could be driven by vision and modified by IVR. As different age groups may be differently affected by IVR, it is necessary to shed light on how age might affect one's interaction with this technology.

Another recent study used IVR to decouple visual information from self-motion and investigate whether adults and 10- and 11-year-old children can optimally integrate visual and proprioceptive cues [29]. An HMD was used to make participants learn a two-legged path either in darkness ("only proprioception"), in a virtual room ("vision + proprioception"), or staying stationary while viewing a pre-recorded video of walking the path in the virtual room ("only vision"). Participants then reproduced this path in darkness. In contrast to what was expected, the authors found that adults failed to optimally integrate visual and proprioceptive cues to improve path reproduction. However, children did integrate these cues to improve their performance. The authors suggest that this may be because children cannot help but rely on visual cues in spatial tasks even when the nature of the task does not require it. We previously

discussed findings demonstrating that HMDs disrupt proprioception, which adults and children rely on in different ways. It may be the case that IVR imparts different effects on adults' and children's performance. We could speculate that, if IVR causes some sort of conflict between vision and proprioception, adults' lack of multisensory integration in these environments could be due to their reliance on proprioception and ability to ignore visual cues. Since this ability to ignore irrelevant visual cues seems not to be mature in children [30], they could benefit from IVR motor training because they would still be using vision to calibrate their less accurate proprioception. It is only recently that the field of IVR research is beginning to focus on the developing child to study developmental differences in relation to their interaction with IVR [31]. Further research is needed to investigate how sensorimotor interaction with IVR changes depending on age-related sensorimotor functioning.

## Statistical approach for exploratory investigations: Bayesian model comparison

Given the lack of evidence concerning the complex interaction between developmental stages, visuo-proprioceptive integration, and IVR, exploratory studies are needed and can benefit from assuming a model comparison approach. Model comparison allows for the selection of the most plausible model given data and a set of candidate models [32]. Firstly, the different research hypotheses are formalized as statistical models. Subsequently, the obtained models are compared in terms of statistical evidence (i.e. support by the obtained data), using information criteria [33]. Information criteria enables the evaluation of models considering the trade-off between parsimony and goodness-of-fit [34]: as the complexity of the model increases (i.e. more parameters), the fit to the data increases as well, but generalizability (i.e. ability to predict new data) decreases. The researchers' aim is to find the right balance between fit and generalizability in order to describe, with a statistical model, the important features of the studied phenomenon, but not the random noise of the observed data.

A Bayesian approach is a valid alternative to the traditional frequentist approach [35, 36], allowing researchers to accurately estimate complex models that otherwise would fail to converge (i.e. unreliable results) in a traditional frequentist approach [37, 38]. Bayesian inference has some unique elements that make the meaning and interpretation of the results different from the classical frequentist approach [39]. In particular, in the Bayesian approach, parameters are estimated using probability distributions (i.e. a range of possible values) and not a single point estimate (i.e. a single value). Bayesian inference has three main components [40]: (1) *Priors*, the probability distributions of possible parameter values considering the information available before conducting the experiment; (2) *Likelihood*, the information given by the observed data about the probability distributions of possible parameter values; (3) *Posteriors*, the resulting probability distributions of possible parameter values, obtained by combining Priors and Likelihood through Bayes' Theorem. As a result, a Bayesian approach assesses the variability (i.e. uncertainty) of parameter estimates and provides associated inferences via 95% Bayesian Credible Intervals (BCIs), the range of most credible parameter values given the prior distribution and the observed data. Thus, a Bayesian approach allows researchers to describe the phenomenon of interest through probabilistic statements, rather than a series of simplified reject/not-reject dichotomous decisions typically used in the null hypothesis significance testing approach [32].

## Research goals and hypotheses

The aim of the present study is to investigate the extent to which the reliability of visual information aids proprioceptive-based self-motion accuracy across the lifespan. We also aim to

explore whether HMD-delivered IVR, compared to equivalent real environments, affects proprioceptive accuracy. Given that findings in the area of multisensory interaction with IVR across development are still conflicting and unexplained with respect to the use of HMDs, the current study seeks to clarify how using an HMD affects children's and adults' self-motion performance, and how these effects could be related to the reliability of the provided visual and proprioceptive information. Research has broadly considered the computer side of IVR features affecting human-computer interaction, but there is a lack of research investigating how individual characteristics of users interact with IVR. To compare performances in reality and IVR, all sensory conditions being equal, would clarify the role of both sensory manipulation and IVR. How might different users, with different levels of multisensory functioning, interact with IVR? The present study explores this question, examining how IVR differs from reality in affecting visuo-proprioceptive integration in adults and children at different developmental stages. Furthermore, the study aims to open new avenues of analysis in this area of research by using a model comparison approach to analyze each hypothesis.

Based on the extant literature described in the introductory section of this work, we hypothesized that children's proprioceptive accuracy would be globally lower than that of adults, but that children would be less impaired than adults by the disruption of proprioception. We further hypothesized that IVR would disrupt proprioception and impact proprioceptive accuracy more in adults than children.

## Materials and methods

### Participants

In order to capture a range of developmental stages, we included primary and secondary school-aged children and adults. We collected data from young children aged from 4 to 8 years old, and older children aged from 9 to 15 years old. This distinction was made to clarify contradictory findings about how long it takes to develop stable proprioceptive accuracy. With regard to the adult group, we included participants within the age range of 18 to 45 years. We excluded older participants based on literature reporting deterioration of proprioceptive accuracy from middle age [41, 42]. For this study, we collected data from 55 participants. In line with our a priori exclusion criteria, we excluded six participants who reported that they had received a diagnosis for any kind of neuropsychological, sensory, or learning disorder from the final analysis. The final sample included 49 participants, distributed across age groups as reported in Table 1.

In a within-subjects design, all participants were exposed to all conditions in a randomized order.

### Materials and set-up

We designed and built a testing room in which different sensory stimulations could be provided and the availability of visual and proprioceptive information could be manipulated while

**Table 1. Participants according to age groups.**

| Age group | | Years | | Range | | Sex | |
|---|---|---|---|---|---|---|---|
| | N | Mean | SD | Min | Max | Male | Female |
| Young Children | 13 | 7.1 | 1.3 | 4 | 8 | 9 | 4 |
| Older Children | 13 | 11.3 | 2.1 | 9 | 15 | 5 | 8 |
| Adults | 23 | 32.4 | 6.7 | 20 | 43 | 12 | 11 |

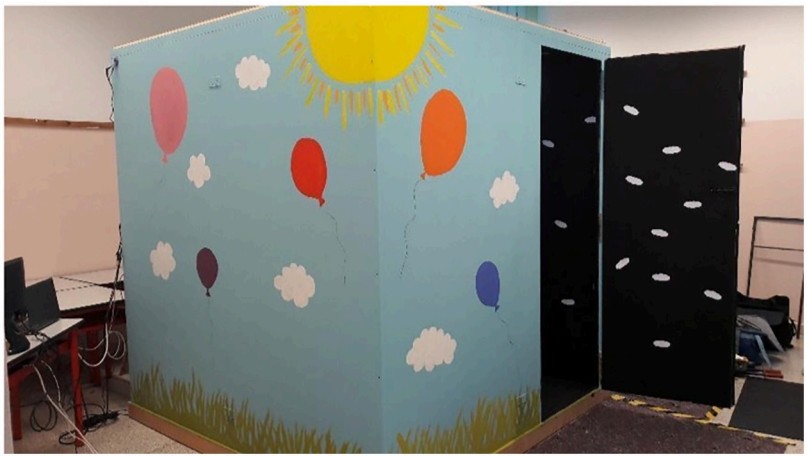

**Fig 1. Experimental room.** The room measured 2 x 2 meters and was soundproof, with black interior walls and equal numbers of white clouds randomly fixed on each wall. The external walls were painted with a child-friendly landscape which was designed to encourage children to enter the room.

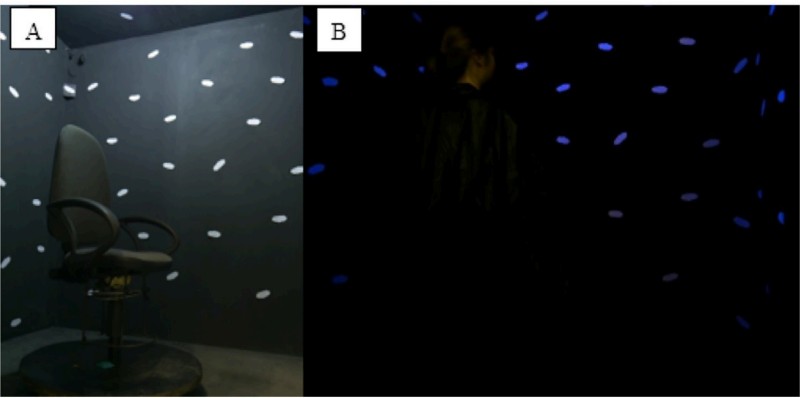

**Fig 2. Experimental room, interior.** A: The swivel chair in a visuo-proprioceptive real environment. B: A participant wearing the black poncho in a vision-only real environment.

completely excluding unwanted external stimuli (Fig 1). In the centre of the room, a customized swivel chair on a round platform was fixed to the floor (Fig 2A). A 360° protractor under the seat was visible to experimenters via a dedicated camera which allowed the measurement of the degree of each rotation. One 50 cm white LED strip (12V DC, 24 Watt per meter) allowed sufficient illumination for a clear and realistic visual experience of the room. One UV lamp (E27 26W) was used to obscure other visual stimuli such that the white clouds on the walls were the only visual cues available. With the UV light on, participants were asked to wear a black poncho which covered their bodies, making them not visible (Fig 2B). One infrared LED spotlight (BIG BARGAIN BW103) enabled clear video recordings of the inside of the room even when it was completely in darkness. This light system was anchored to the ceiling, over participants' heads, and was covered by a black panel which prevented participants from directly seeing the lights.

We provided the IVR simulation through the HMD Oculus Gear VR 2016, 101° FOV, 345 g weight, interfaced with a Samsung Galaxy S7 (ANDROID 8.0.0 operating system).

A Nikon KeyMission 360 camera was used to create 360˚ images of the room and to build the IVR. The room was monitored via one USB 2.0 DirectShow webcam, and one USB 2.0 DirectShow webcam with integrated infrared LED.

To monitor the video recordings and IVR simulations, we used a SATELLITE Z30-B, Windows 10, 64bit, Intel Core i5-5200U CPU @ 2.20 Ghz, 8.0 GB RAM, Intel HD Graphics 5500. The communication between people inside and outside the room was enabled via a system of a USB speaker, microphone, headphones, and one USB soundcard. The VR server application developed for this experiment is an Android application with VR environments, developed in Unity. A remote interface, also developed in Unity for Windows or Android OS, allowed experimenters to control the VR server application. Software for audio-video recording and real-time communication was developed in TouchDesigner.

## Procedure

Adult participants were welcomed into the lab and asked to sign a consent form. Parents of children were asked to sign the form on their child's behalf. The study was approved by the Ethics Committee of Psychology Research, University of Padua. At least two experimenters conducted the experiment. On commencing the experiment, participants were asked to sit on the swivel chair which was fixed in the middle of the recording area inside the room. Experimenter 1 would close the door and stay inside near the participant for the duration of the experiment. Experimenter 2 managed the experiment from outside the room: they switched the lights on and off, changed the visual stimuli which were presented through the HMD, and gave verbal instructions to Experimenter 1 and to the participants. Although the room is soundproof, Experimenter 2 could communicate with the people inside through a microphone and speaker system. During the experimental task, Experimenter 1 managed the passive rotation and remained silent behind the participant, providing no visual or auditory cues.

## Experimental task

We adopted a self-turn paradigm in which the experimenter rotates the chair a certain degree (passive rotation) from a *start position* to an *end position*. After each passive rotation, participants were asked to rotate back to the start position (active rotation). The position at which the participant stopped their active rotation is recorded as the *return position*. During the passive rotation, participants sat still and kept their feet on a footrest which rotated with the chair. To perform the active rotations, participants could use their feet on the still platform under the chair to move themselves. Within a given experimental condition, during both the encoding *(passive rotation)* and the recall *(active rotation)* phase, all sensory information was consistent. During the recall phase, proprioception derived from the active movement was involved in performing the *active rotation* and recalling the *start position*. This constitutes the accuracy measure in our task, in line with the extant literature [43–45]. We did not manipulate vestibular information, which was consistent across all experimental conditions. On the other hand, we manipulated vision across the three experimental conditions as described in the following section.

## Conditions

The experiment had a multifactorial design with one between-subjects factor (Age) and four within-subjects independent variables (Environment, Perception, Amplitude, Direction). Therefore, we had a 3 (young-children/older-children/adults) x 2 (Reality/IVR) x 3 (Proprioception/Vision/Vision+Proprioception) x 2 (clockwise/counterclockwise) design, with an additional continuous independent variable of rotation amplitude. Within the environment

variable, there were reality conditions in a real environment (the interactive room) and IVR conditions with participants wearing the HMD that showed 360˚ pictures of perceptually equivalent versions of the reality conditions. Within the perception variable, there were three conditions. One blind condition removed all visual information such that only proprioceptive information could be used (P). One visual condition limited the access to proprioceptively informative visual landmarks (hiding the participants' body and the room corners) in order to disrupt proprioception, while providing a proprioceptively uninformative visual texture (a pattern of small bright clouds on the walls) (V). Indeed, previous research has found that after being disorientated by a passive rotation in a real environment, people could still detect the position of global landmarks (the room's corners), while making huge errors locating surrounding objects [7]. Our intention was to disrupt proprioception through altering the visual information available, without making changes to the proprioceptive information arising from participants' bodies during the passive and active movements, which are consistent within participants. The last condition allowed the participant to access reliable visual and proprioceptive information (VP). We aimed to check whether the equivalent visual information would lead to equivalent proprioceptive accuracy when comparing reality and IVR conditions. In fact, the degree to which visual cues aid proprioception seems to be environment-specific. For instance, in HMD-delivered IVR, users' self-motion could not benefit so much from global landmarks [46]. Although it was not a main aim of the experiment, we aimed to control whether the rotation direction and amplitude would affect performance. For this purpose, the passive rotation of each condition was made in both directions (clockwise—"R", counterclockwise—"L"), and with two angle amplitudes (90 and 180 degrees). As the passive rotation was manually performed by the experimenter, perfect accuracy in reaching 90 and 180 degrees was not possible. Given the variability in the actual passive rotations, we considered amplitude as a continuous variable. In this way, we controlled for this potential source of noise. The order of conditions was randomized. Participants performed two trials per Environment X Perception condition, resulting in 12 observations per participant.

The experimental conditions are as follows:

1. R_P (Reality; only proprioception: no visual information available).

2. R_V (Reality; only vision: proprioceptively uninformative visual texture of small bright clouds on the walls. No first-person view of the body or room corners in order to disrupt proprioception by manipulating vision).

3. R_VP (Reality; proprioceptively informative visual cues available, including first-person view of the body and room corners. The visual texture of clouds on the walls is available).

4. IVR_P (HMD on; only proprioception: no visual information available).

5. IVR_V (HMD on; only vision: proprioceptively uninformative visual texture of small bright clouds on the walls. No first-person view of the body or room corners in order to disrupt proprioception by manipulating vision).

6. IVR_VP (HMD on; proprioceptively informative visual cues available, including visible room corners, although the first-person view of the body is not visible. The visual texture of clouds on the walls is available).

## Measures of task performance

The proprioceptive accuracy of self-turn performances was calculated in terms of error as the absolute difference between the *start position* (from which the experimenter started the passive

rotation) and the *return position* (in which the participant stopped the active rotation). In this way, greater values indicated a less accurate performance, where a value of 0 would indicate that the participant actively rotated back to the exact start position, and a value of 100 would indicate that the participant actively rotated back to a position that was 100 degrees away from the start position.

Proprioceptive accuracy was manually measured during an offline coding of the video recording. The video shows two matched recordings of both the entire room (with the participant and Experimenter 1 in frame) and the protractor positioned under the seat of the swivel chair. A vertical green line was superimposed on the protractor image to facilitate detection of the specific degree of each rotation. Two independent evaluators coded the videos and entered the start and return positions in the dataset. Values which were divergent for more than two degrees were a priori considered disagreement values. That was the case for 82 out of 578 observations (14.2%). A third coder examined the video recordings of the disagreement values to make the final decision. In case of a disagreement value, the third coder's value was used instead of the value that differed most from the third coder's value. We obtained a dataset with two codings for each piece of data. We evaluated the intercoder agreement by conducting an intra-class correlation (ICC), which is one of the most commonly used statistics for assessing inter-rater reliability (IRR) for ratio variables [47]. On the double values indicating the start, end, and return positions of each rotation, the ICC index has been calculated. The analysis estimates an ICC = .99. This nearly perfect inter-coder agreement derives from the small mean difference between the two coders' values (Mean$_{coderA-coderB}$ < .16), within the huge range of possible values (0/360). We carried out the data analysis on the final dataset with the average of the two values.

## Statistical approach

In order to explore how Age, Perception conditions, and Environment conditions interact to affect proprioceptive accuracy, a model comparison approach was used. Firstly, each research hypothesis was formalized as a statistical model. Subsequently, the obtained models were compared in terms of statistical evidence (i.e. support by the data) using information criteria [33].

Given the complex structure of the data, Bayesian generalized mixed-effects models were used [35, 48]. Specifically, data were characterized by: (1) a continuous non-normally distributed dependent variable (i.e. rotation error); (2) a between-subject factor (i.e. Age); (3) within-subject factors (i.e. Perception condition and Environment condition); (4) a quantitative independent variable (i.e. rotation Amplitude). Mixed-effects models allow us to take into account the repeated measures design of the experiment (i.e. observations nested within participants). Thus, participants were treated as random effects, with random intercepts that account for interpersonal variability, while the other variables are considered as fixed effects. Generalized mixed-effects models were used considering the Gamma distribution, with logarithmic link function, as the probability distribution of the dependent variable. Generalized mixed-effects models allow us to model non-normally distributed data using appropriate probability distributions that reflect the characteristics of the data [49]. Selecting an appropriate probability distribution provides better fit to the data and more reliable results [50]. Gamma distribution is advised in the case of positively skewed, non-negative data, when the variances are expected to be proportional to the square of the means [51]. These conditions are respected by our dependent variable: we only have positive values, with a positive skewed distribution, and we expect a greater variability of the possible results as the model predicted mean increases (i.e. a greater dispersion of participants' scores when greater mean values are predicted by the model).

Analyses were conducted with the R software version 3.5.1 [52]. Models were estimated using the R package *'brms'* [53] which is based on STAN programming language [54, 55] and

employs the No-U-Turn Sampler (NUTS; [56]), an extension of Hamiltonian Monte Carlo [57]. All our models used default prior specification of the R package *'brms'* [53]. Detailed prior specifications are reported in the Supplemental Materials. These priors are considered non-informative since they leave the posterior distributions to be mostly influenced by the observed data rather than by prior information. Each model was estimated using 6 independent chains of 8,000 iterations with a "warm-up" period of 2,000 iterations, resulting in 36,000 usable samples.

Convergence was evaluated via visual inspection of the trace plots (i.e. sampling chains) and R-hat diagnostic criteria [58]. All tested models showed satisfactory convergence with all R- hat ≤ 1.0008, where values close to 1 indicate convergence, and none exceed the 1.100 proposed threshold for convergence [35]. All R-hat values and trace plots are reported in the Supplemental Materials.

The Watanabe-Akaike information criterion (WAIC; [59, 60]) was used as information criteria to select the most plausible model among the tested models, given the data. WAIC is the corresponding Bayesian version of the commonly used Akaike information criterion (AIC; [61]). WAIC weights were computed to present the probability of each model of making the best predictions on new data, conditional on the set of models considered [32]. This allows for the comparison of models with a continuous informative measure of evidence. Finally, the most plausible model was interpreted considering the estimated posterior parameter distributions. Main effects and interaction effects were evaluated using planned comparison and graphical representations of the predicted values by model.

The full analysis report is available in the S1 Appendix.

## Results

### Descriptives

Out of the 49 participants, 43 participants completed the task in all 12 trials, 4 participants completed 11 trials, 1 participant completed 10 trials, and 1 participant completed 8 trials. This failure to complete all trials with some participants was due to technical problems which occurred with the experimental apparatus. Thus, the final data consist of 578 observations nested in 49 participants. The number of observations in each condition is reported in S1 Table.

We considered Amplitude of the passive rotations as a continuous variable whose distribution is shown in Fig 3. To obtain interpretable results in the analyses, the Amplitude variable was standardized (i.e., Z scores were obtained).

The mean self-turn error in the present sample was 17.1 degrees (SD = 8.0). The frequency of the observed values is reported in Fig 4. Considering how we computed the self-turn error, only positive values are possible and from visual inspection, the dependent variable has an evident positive skewed distribution.

The means and standard deviations of the self-turn error for the three age groups in the six different experimental conditions are reported in Table 2 and the distributions of the observed data are presented in Fig 5. For the sake of interpretability, descriptive statistics were computed according to Age, Environment, and Perception, without taking into account the variable Amplitude (i.e., all observations in the same condition were considered independently of the Amplitude values), which will be considered later on in the analysis. Considering the observed values according to Age, adults (M = 12.8, SD = 4.4) made less self-turn errors than older children (M = 16.4, SD = 7.5) and young children (M = 25.3, SD = 7.7). Looking at the Environment conditions, participants made less errors and were thusly more accurate in the reality condition (M = 13.9, SD = 8.0) than in the IVR condition (M = 20.2, SD = 10.3). Finally,

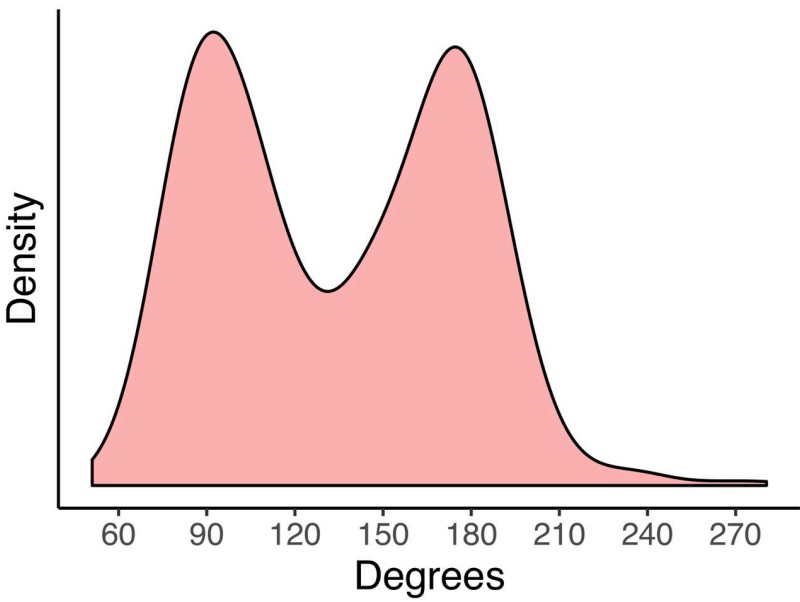

**Fig 3. Estimated distribution of the actual amplitude in the passive rotation.** ($n_{participants}$ = 49; $n_{observations}$ = 578).

considering the different levels of the variable Perception, participants made less self-turn errors when they could rely on both vision and proprioception (M = 13.9, SD = 11.3) than when they could use only vision (M = 14.5, SD = 9.3) or proprioception (M = 22.8, SD = 14.1).

## Model comparison and interpretation

Seven different Bayesian generalized mixed-effects models were performed to analyze the data (see S2 Table). In each model the dependent variable was the error in the self-turn task. WAIC

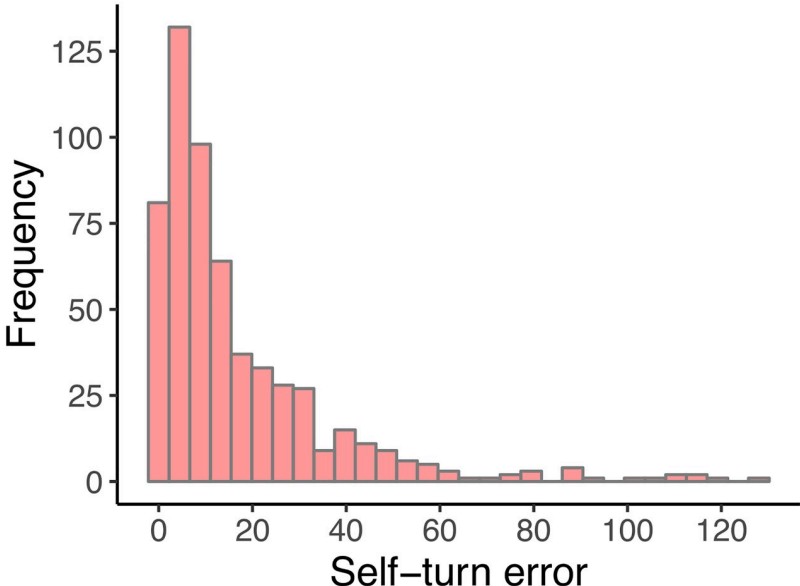

**Fig 4. Frequencies of the observed self-turn errors.** ($n_{participants}$ = 49; $n_{observations}$ = 578).

**Table 2. Descriptive statistics.** Means and standard deviations of self-turn error according to age and the experimental conditions.

| | Perception | | | | | | Total | |
| | Proprioception | | Vision | | Vision + Proprioception | | | |
| | Mean | SD | Mean | SD | Mean | SD | Mean | SD |
|---|---|---|---|---|---|---|---|---|
| **Reality** | | | | | | | | |
| Adults | 16.2 | 8.6 | 9.8 | 12.6 | 6.1 | 4.1 | 10.7 | 6.0 |
| Older Children | 19.6 | 10.5 | 14.0 | 18.2 | 6.7 | 3.6 | 13.5 | 7.3 |
| Young Children | 30.6 | 22.4 | 8.2 | 5.2 | 20.7 | 20.5 | 19.8 | 9.0 |
| Total | 20.9 | 15.0 | 10.5 | 12.9 | 10.1 | 12.5 | 13.9 | 8.0 |
| **IVR** | | | | | | | | |
| Adults | 17.6 | 10.6 | 13.5 | 7.6 | 13.7 | 9.1 | 14.9 | 6.3 |
| Older Children | 23.6 | 19.1 | 17.5 | 10.1 | 16.9 | 18.6 | 19.3 | 9.5 |
| Young Children | 37.8 | 16.2 | 28.5 | 16.5 | 25.1 | 16.5 | 30.3 | 9.9 |
| Total | 24.7 | 16.8 | 18.5 | 12.6 | 17.4 | 14.6 | 20.2 | 10.3 |
| **Total** | | | | | | | | |
| Adults | 17.1 | 6.4 | 11.8 | 8.0 | 9.9 | 4.8 | 12.8 | 4.4 |
| Older Children | 21.6 | 13.7 | 15.7 | 11.8 | 11.7 | 9.4 | 16.4 | 7.5 |
| Young Children | 34.2 | 18.0 | 18.2 | 7.9 | 23.4 | 15.8 | 25.3 | 7.7 |
| Total | 22.8 | 14.1 | 14.5 | 9.3 | 13.9 | 11.3 | 17.1 | 8.0 |

*Note:* $n_{participants}$ = 49; $n_{observations}$ = 578.

results indicated that m.2 was the most plausible model for the observed data. It evaluated the 2-way interaction effect between Perception and Environment conditions, and had the lower WAIC value (WAIC = 4345.3) and a probability of being the best of.67. WAIC values and relative WAIC weights of all models are reported in S3 Table.

In order to interpret the effects of model m.2, 95% Bayesian Credible Intervals (BCI) of the parameters posterior distribution were evaluated (S4 and S5 Tables). Ninety-five percent BCI represent the range of the 95% most credible parameters values given the prior distribution and the observed data. Thus, an effect is considered plausible if the value zero is not included in the 95% BCI, whereas if the value zero is included in the 95% BCI, it is interpreted as not plausible.

Self-turn error was moderated by Amplitude, by Age, and by the interaction between Perception and Environment conditions. On the contrary, the direction of rotations seems to have no effect on the participants' performance ($\beta$ = .10; 95% BCI = -.04; .23).

To evaluate the model fit (i.e. the model's ability to explain the data) we used a Bayesian definition of R-squared [62] to estimate the proportion of variance explained. The estimated value of Bayesian R-squared for the model m.2 is.26 (95% BCI = .19; .34), that is the model explains 26% of the variability of the data.

**Rotation amplitude.** Self-turn error was moderated by Amplitude ($\beta$ = .22; 95% BCI = .14; .29), for which increasing rotation amplitude is associated with a worse performance (Fig 6).

**Group age.** To evaluate the role of Age, the distributions of predicted mean values for the three groups were considered (Fig 7). BCI values are reported in S4 Table. The predicted mean error for adults was 12.8 degrees (95% BCI = 10.6; 15.1), for older children was 15.5 degrees (95% BCI = 12.1; 19.2) and for young children was 24.8 degrees (95% BCI = 19.3; 30.8). Bayesian pairwise comparisons (i.e., predicted score differences between groups) showed that overall, young children are expected to make more self-turn errors than adults (95% BCI = 6.3;

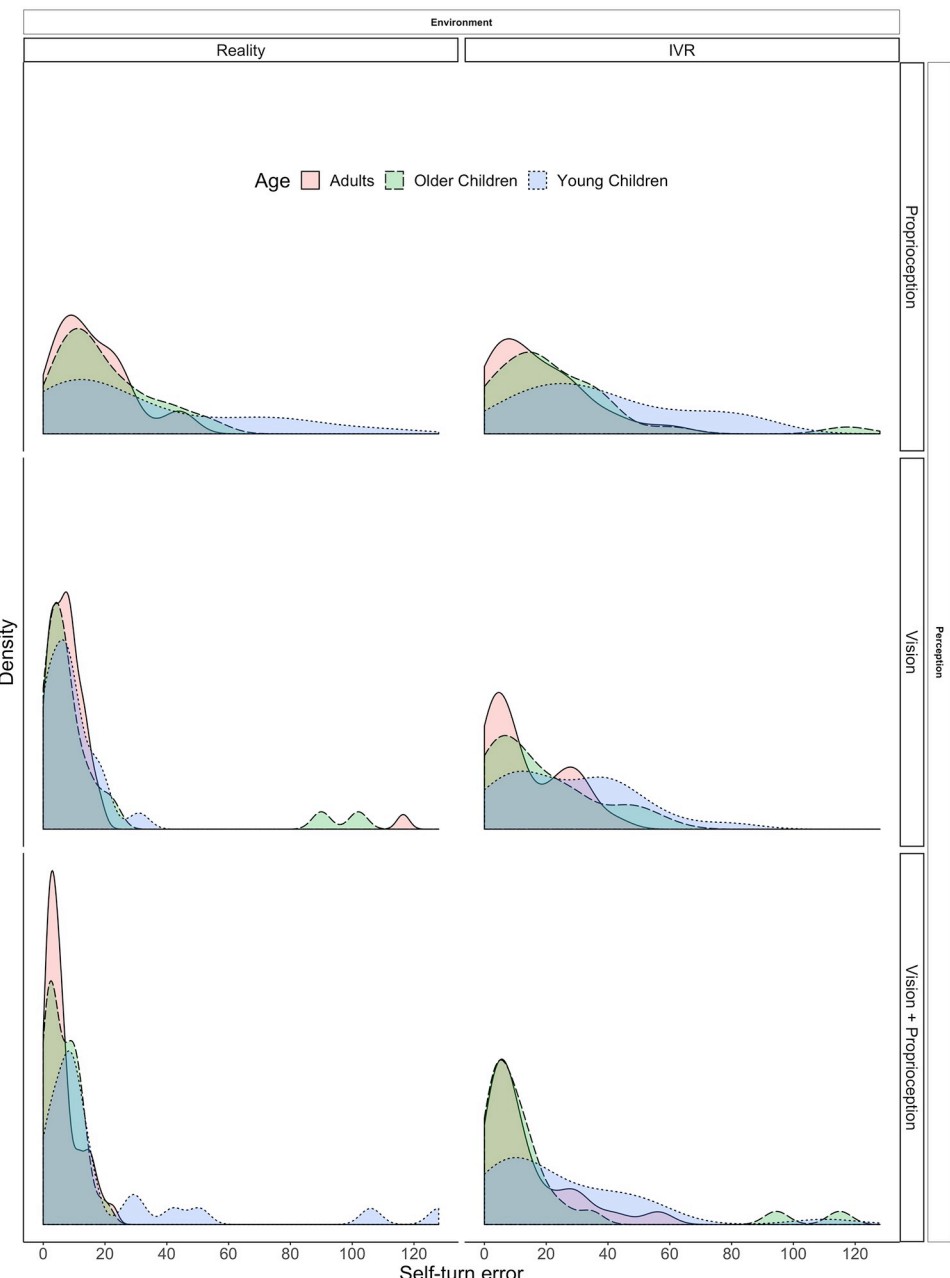

**Fig 5. Estimated distributions of the observed self-turn errors in the different conditions according to age.** ($n_{participants}$ = 49; $n_{observations}$ = 578).

18.2) and also more than older children (95% BCI = 2.8; 16.0). However, we cannot state that older children are expected to make more self-turn errors because the 95% BCI of the difference includes the value zero (95% BCI = -1.4; 6.9).

**Perception and environment.** To interpret the interaction between the Perception and Environment conditions, the distributions of predicted mean values for all six conditions were considered (Fig 8). BCI values are reported in S5 Table. In the Reality conditions, the predicted mean error for proprioception was 22.4 degrees (95% BCI = 18.0; 27.2), for vision was 11.3 degrees (95% BCI = 9.0; 13.9) and for vision + proprioception was 9.8 degrees (95% BCI = 7.8;

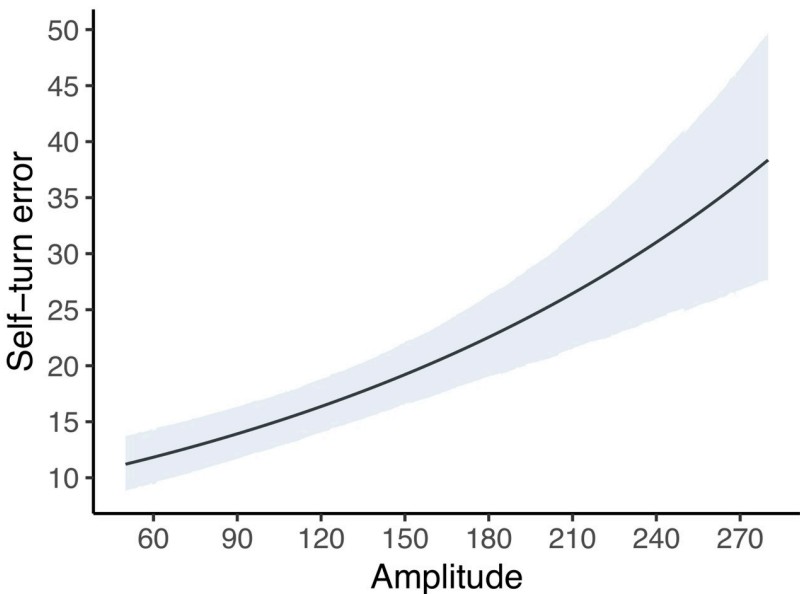

**Fig 6. Predicted mean of self-turn error according to amplitude ($n_{participants}$ = 49; $n_{observations}$ = 578).** The line represents the mean value, the shaded area the 95% BCI values.

12.0). In the IVR conditions, the predicted mean error for proprioception was 24.3 degrees (95% BCI = 19.3; 29.2), for vision was 18.0 degrees (95% BCI = 14.4; 21.8) and for vision + proprioception was 17.8 degrees (95% BCI = 14.2; 21.7). Bayesian pairwise comparisons (i.e predicted error differences between conditions) showed that in both Reality and IVR, participants are expected to make more self-turn errors when they rely only on proprioception than when they can use only vision (Reality: 95% BCI = 6.5; 15.8; IVR: 95% BCI = 0.9; 11.7) or vision

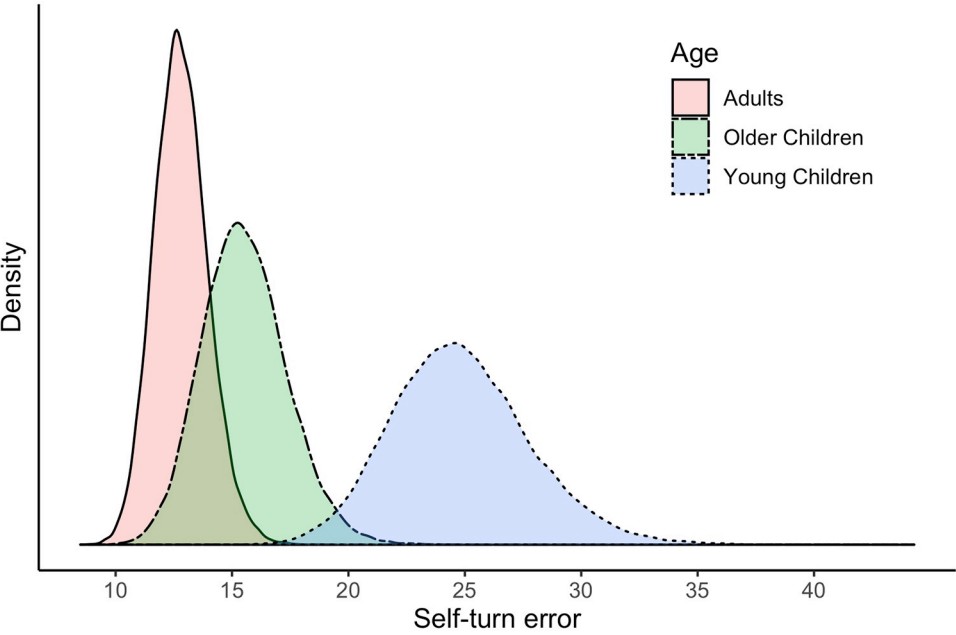

**Fig 7. Distributions of the predicted means of self-turn error according to age.** ($n_{participants}$ = 49; $n_{observations}$ = 578).

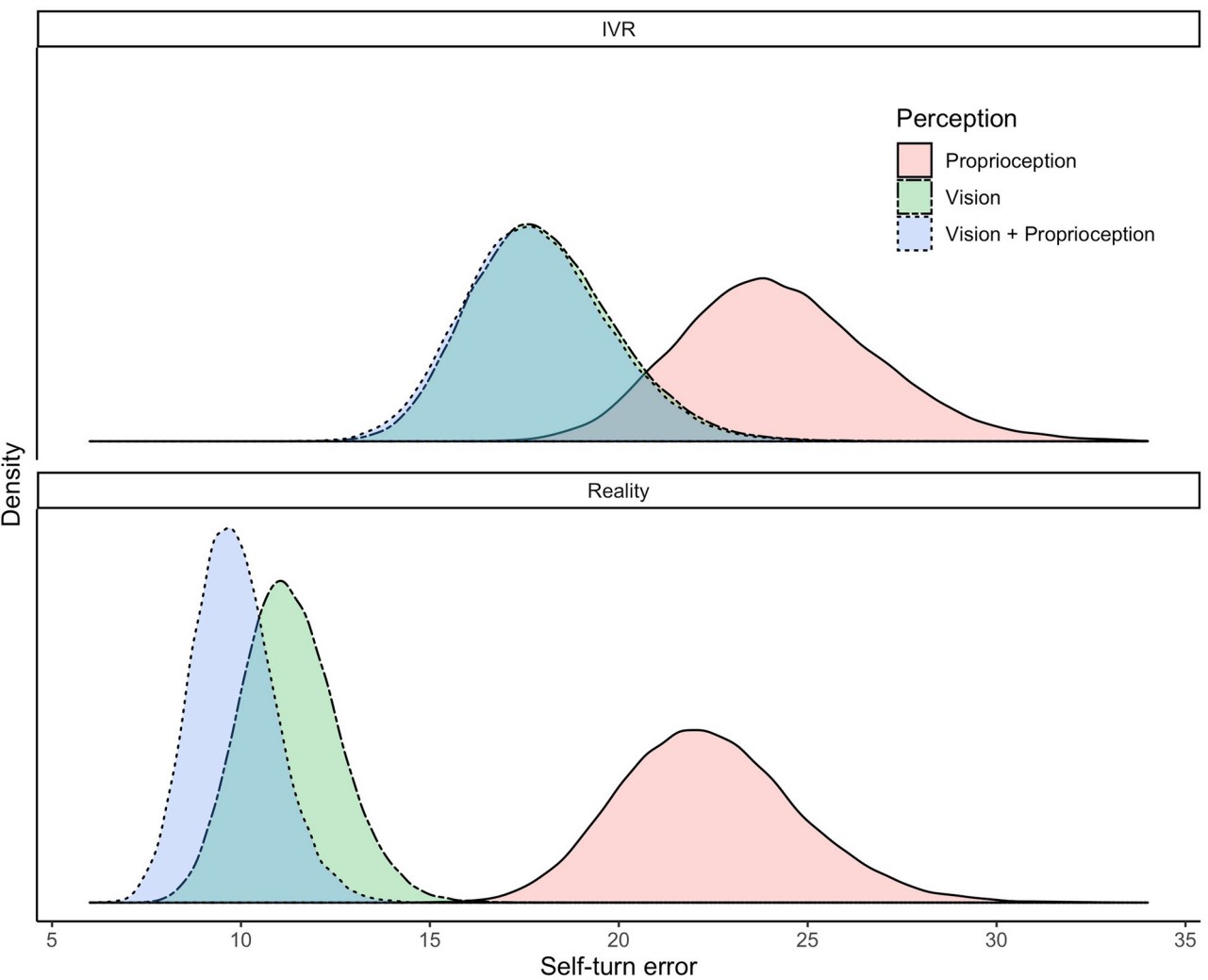

**Fig 8. Distributions of the predicted means of self-turn error according to the different conditions.** ($n_{participants}$ = 49; $n_{observations}$ = 578).

+ proprioception (Reality: 95% BCI = 8.0; 17.2; IVR: 95% BCI = .08; 11.7). In addition, in both environments there is no difference between the use of vision and vision + proprioception (Reality: 95% BCI = -1.4; 4.4; IVR: 95% BCI = -4.3; 4.9). Moreover, comparing IVR to Reality conditions, results show that while wearing the HMD the self-turn errors increase when participants rely only on vision (95% BCI = 2.8; 10.6) or on vision + proprioception (95% BCI = 4.3; 11.9). On the other hand, participants are not expected to make more errors than in Reality when they rely only on proprioception (95% BCI = -4.3; 7.5).

**Effect size.** To quantify the differences between the various age groups and conditions, we expressed the effects as the ratio between the two scores of the comparison of interest (see S6 Table). Thus, for example, young children are expected to make 88% more errors than adults and 58% more errors than older children. Considering the Reality environment conditions, when using only proprioception participants are expected to make 92% more errors than when they rely only on vision and 118% more errors than when using vision + proprioception. Considering the IVR conditions, when using only proprioception participants are expected to make 34% more errors than when they rely only on vision and 35% more errors than when

using vision + proprioception. Moreover, comparing IVR to the Reality condition, in IVR participants are expected to make 56% more errors when using only vision and 75% more when using vision + proprioception.

## Discussion

This experiment explored the extent to which visual information aids proprioceptive-based self-motion accuracy across the lifespan, and specifically in three developmental groups: 4–8-year-old children, 9–15-year-old children, and adults. Moreover, the experiment assessed whether HMD-delivered IVR affects accuracy.

As expected, we found a main developmental trend in the improvement of proprioception across conditions. In particular, as hypothesized, we found differences between the young child group (4–8 years old) and the older child and adult groups (9–15 and 20–43 years old), with this youngest group showing lower proprioceptive accuracy than the two older groups. This indicates that proprioceptive development predominantly takes place in the first eight years of life, such that adolescent and pre-adolescent children make more accurate proprioceptive judgements than younger children.

In line with our hypotheses, we also found an interaction effect between Perception and Environment. Our findings indicate that proprioceptive accuracy was markedly impaired when participants could rely only on proprioceptive input, regardless of the environment. In the conditions which forced participants to rely solely on proprioception by removing all visual information, all groups were less accurate than in conditions where visual information was provided, regardless of the proprioceptive salience of this visual information. This finding is consistent with the assertion that visual and vestibular information combine with proprioceptive information to allow accurate self-motion [9]. Moreover, it indicates that typically developing child and adult populations rely specifically on vision to calibrate proprioception in order to accurately judge their movements. Regarding the role of different visual landmarks, no differences were found between vision + proprioception and vision only conditions, that is, conditions in which participants could view all aspects of the real or virtual room versus conditions in which participants only saw a visual texture of randomly placed clouds but were unable to see proprioceptively informative visual cues such as the corners of the room or their body. Moreover, IVR, compared to Reality, disrupted proprioception only when visual input was provided (vision + proprioception and vision only conditions). There were no differences between IVR and Reality in only proprioception (blind) conditions. This allows us to exclude the possibility that wearing the HMD alone, and the corresponding weight and head restriction, might have disrupted proprioception. We did find that performance worsened in IVR conditions where visual information was available relative to corresponding reality conditions. The way in which the HMD delivers visual information has a complex (and essentially unknown) effect on self-motion perception and the kinematics of movement [63]. Factors such as display type, field of view, visual content (peripheral cues, high-low visual contrast, etc.), temporal lag between the user's action and the HMD's reaction, and so on could be the means by which IVR disrupts proprioception through vision. This is an important finding, given that few IVR experiments have considered that performance may be affected simply due to the use of HMD-delivered IVR. Many previous IVR experiments seem to implicitly assume that performance in IVR constitutes an appropriate corollary for real-world performance, but our findings indicate that this may not be the case. Despite this HMD effect, our results provide evidence that IVR may be a useful means of studying multisensory integration and accuracy. Indeed, the same general Perception trend in self-motion accuracy (proprioception only, vision only, vision + proprioception) was found both in IVR and Reality environments.

In contrast to our expectations, we failed to find any Age x Perception interaction effect. We expected that adults would be more affected by disrupted proprioception than children, but this was not the case. Various aspects of the experimental design should be taken into account to discuss this result. Firstly, our manipulation of the multisensory input in different conditions could have been insufficient to uncover the expected differences. We found the expected general trend of reduced proprioceptive accuracy in vision conditions relative to vision + proprioception conditions. However, this difference failed to reach a meaningful magnitude. As previous studies highlight, relative dominance of visual and proprioceptive input and visuo-proprioceptive integration are task-dependent [2, 26]. For example, proprioception has been reported to be more precise in the radial (near-far) direction and vision in the azimuthal (left-right) direction [64–66]. It could be suggested that our azimuthal proprioceptive task was too dependent on vision to allow the detection of differences that were due to the disruption of proprioception. In fact, our "only vision" conditions were designed to disrupt proprioception by removing proprioceptively informative visual cues (the room corners and participant's body), while still providing proprioceptively uninformative visual landmarks (surrounding texture of clouds). It could be the case that proprioceptively uninformative visual landmarks are sufficient to allow accurate performance in our task. In addition, we based our research on similar studies that used a standing self-turn paradigm [7, 67]. We utilized a seated self-turn paradigm so that we could use the chair position as a precise and consistent measurement point of reference, independently from the participants' individual postures which may vary. However, this seated task could be less challenging than a standing one, resulting in a ceiling effect, particularly for older children and adult groups. Moreover, we failed to find any Age x Environment interaction, which prevents us from providing evidence on age-dependent user-IVR interactions. Increased knowledge in this area could have meaningful implications for fields such as IVR education, rehabilitation, and therapy, shedding light on when and how IVR interventions could be effective at different developmental stages. Future research could focus specifically on children younger than eight years old to explore the early development of visuo-proprioceptive integration, as well as potentialities and threats related to IVR use.

We also found a main effect of rotation Amplitude, with proprioceptive accuracy consistently decreasing as rotation amplitude increased. It is possible that this effect is specifically due to working memory constraints [43, 44]. In our task, accuracy largely depends on participants' ability to actively maintain the start position in memory, and it may be the case that differences in working memory capacity across age groups and conditions could have affected results. As the study of the effect of rotation amplitude was not a primary goal of this work, we did not explore interaction effects between Amplitude and other variables (i.e. Age, Perception, or Environment). Remarkably, working memory limitations have been found up to pre-adolescence [68] and age-related lower visuo-spatial working memory capacity can be associated with lower proprioceptive accuracy in body position-matching tasks [69]. A more in-depth look is also necessary to investigate potential implications of both the proprioceptive and visual sensory register and its influence on performance, as individual sensory registers have been shown to affect working memory in multisensory environments (for a review, see [70]).

The present study opens intriguing perspectives for future research, despite having some limitations. Firstly, the experimenter manually rotated the participant, so although experimenters were trained to keep a similar speed and method of rotating, the rotation velocity was not perfectly consistent across trials and participants, potentially influencing participants' performance as in previous research [67]. Another limitation concerned the manipulation of visual conditions distinguishing between "only vision" and "vision + proprioception". As we found no meaningful differences between these two Perception conditions, the "only vision"

condition could have been insufficient to isolate vision and disrupt proprioception as we aimed to. It would be interesting to see how similar but more effective manipulations of visual information aimed at disrupting proprioception would affect performance. Moreover, the age groups could be too broad to clearly show early developmental trends and changes.

One of the most intriguing yet unexplored perspectives that led to this work concerns the possibility of intentionally disrupting proprioception through HMD-delivered IVR. This method could be employed to study the degree to which different developmental populations rely on proprioception, vision, and visuo-proprioceptive integration. From an applied perspective, disrupting proprioception could comprise an innovative intervention for use with clinical populations which demonstrate an atypical reliance on specific senses and atypical integration of vision (*exteroception*) and proprioception. For example, people with Autism Spectrum Disorder (ASD) seem to show an over-reliance on proprioception and hypo-reliance on vision [71–73]. This perceptual strategy might not only lead to impaired motor skills in ASD (e.g. dyspraxia and repetitive behaviors), but also seems to be related to core features of impaired social and communicative development. Interventions could be aimed at increasing the reliance on vision in children with ASD by disrupting proprioception. In this respect, a possible speculation is that IVR interventions could constitute a useful training method to achieve a therapeutic purpose.

## Conclusion

In sum, the present study offers useful insights regarding the use of IVR in research on multisensory integration and sensorimotor functioning. When visual information is provided, proprioceptive accuracy in IVR seems to be impaired relative to performance in reality. As proprioception is fundamental to performance in any motor task, this has to be taken into account when interpreting the results of IVR studies which involve proprioceptive abilities. However, IVR could still be a useful tool for detecting multisensory trends. In fact, we found the same condition-specific trend in IVR as in reality. Both in reality and IVR, the conditions which allowed a reliance solely on proprioception led to the lowest proprioceptive accuracy, and minimal differences emerged between vision only and vision + proprioception conditions. The exploratory nature of the present study could contribute to the undertaking of more confirmatory future studies, which would benefit from the estimated effect sizes provided here, to develop and test further hypotheses.

## Supporting information

**S1 Appendix. Full analysis report.**
(PDF)

**S1 Table. Frequency of self-turn errors.**
(PDF)

**S2 Table. Model formulas.**
(PDF)

**S3 Table. WAIC model comparison.**
(PDF)

**S4 Table. Predicted means and differences of self-turn error according to age.**
(PDF)

**S5 Table. Predicted means and differences of self-turn error according to experimental conditions.**
(PDF)

**S6 Table. Effect size as the ratio of the scores of the different age groups or experimental conditions.**
(PDF)

## Acknowledgments

Our gratitude to the multimedia designers Marco Godeas and Carlo Marzaroli. They did a great job in building the IVR and setting up the laboratory. This research was helped immensely by their experience with IVR technologies and children with Autism Spectrum Disorder. This experiment would not have been possible without their ideas and technical support.

Thanks to F.lli Budai S.r.l. for building the experimental room: a generous gift for which we are very grateful.

Thanks to Beneficentia Stiftung Foundation for supporting our research.

Sincere thanks to Associazione Pro Musica Ruda—Scuola Comunale di Musica and the Comune di Ruda (Udine, Italy) for hosting our laboratory and supporting our work.

## Author Contributions

**Conceptualization:** Irene Valori, Phoebe E. McKenna-Plumley, Rena Bayramova, Teresa Farroni.

**Data curation:** Irene Valori, Phoebe E. McKenna-Plumley, Rena Bayramova, Claudio Zandonella Callegher.

**Formal analysis:** Irene Valori, Claudio Zandonella Callegher, Gianmarco Altoè.

**Funding acquisition:** Teresa Farroni.

**Investigation:** Irene Valori, Phoebe E. McKenna-Plumley, Rena Bayramova, Teresa Farroni.

**Methodology:** Irene Valori, Phoebe E. McKenna-Plumley, Rena Bayramova, Teresa Farroni.

**Project administration:** Irene Valori, Teresa Farroni.

**Resources:** Teresa Farroni.

**Supervision:** Teresa Farroni.

**Visualization:** Irene Valori, Phoebe E. McKenna-Plumley, Rena Bayramova, Claudio Zandonella Callegher, Gianmarco Altoè.

**Writing – original draft:** Irene Valori, Phoebe E. McKenna-Plumley, Rena Bayramova, Claudio Zandonella Callegher, Teresa Farroni.

**Writing – review & editing:** Irene Valori, Phoebe E. McKenna-Plumley, Rena Bayramova, Claudio Zandonella Callegher, Gianmarco Altoè, Teresa Farroni.

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
