## [Decision Letter · Decision Letter 0]

24 Oct 2019

PONE-D-19-23517

Proprioceptive accuracy in immersive virtual reality: A developmental perspective

PLOS ONE

Dear Associate professor Farroni,

Thank you for submitting your manuscript to PLOS ONE. After careful consideration, we feel that it has merit but does not fully meet PLOS ONE’s publication criteria as it currently stands. Therefore, we invite you to submit a revised version of the manuscript that addresses the points raised during the review process.

The reviewers overall evaluated positively the novelty of finding and robustness of results. At the same time, they raised numbers of constructive comments. Among them two issues seemed crucial for me to improve the paper. Firstly, authors should further clarify the method being used, and evaluate any potential source of noise or lack of control. For example, several reviewers raised concern about the lack of control in the procedure to rotate the chair. Secondly, authors should clarify the text overall, particularly for the consistency in terminology. Reviewers also found the text sometimes too long and difficult to follow. I recommend the authors to revise, and make it more concise and clear. I also recommend the authors to reply all the other constructive comments from the reviewers. 

I will send the revised manuscript to the original reviewers and ask them to evaluate whether you have fully addressed their concerns.

We would appreciate receiving your revised manuscript by Dec 08 2019 11:59PM. To enhance the reproducibility of your results, we recommend that if applicable you deposit your laboratory protocols in protocols.io, where a protocol can be assigned its own identifier (DOI) such that it can be cited independently in the future. For instructions see: http://journals.plos.org/plosone/s/submission-guidelines#loc-laboratory-protocols

We look forward to receiving your revised manuscript.

Kind regards,

Atsushi Senju

Academic Editor

PLOS ONE

**Journal Requirements:**

**Comments to the Author**

1. Is the manuscript technically sound, and do the data support the conclusions?

Reviewer #1: Partly

Reviewer #2: Yes

Reviewer #3: Yes

Reviewer #4: Yes

2. Has the statistical analysis been performed appropriately and rigorously? 

Reviewer #1: Yes

Reviewer #2: Yes

Reviewer #3: Yes

Reviewer #4: I Don't Know

3. Have the authors made all data underlying the findings in their manuscript fully available?

Reviewer #1: Yes

Reviewer #2: Yes

Reviewer #3: Yes

Reviewer #4: Yes

4. Is the manuscript presented in an intelligible fashion and written in standard English?

Reviewer #1: Yes

Reviewer #2: Yes

Reviewer #3: Yes

Reviewer #4: Yes

5. Review Comments to the Author

Reviewer #1: This study investigated the developmental process of proprioceptive ability and the impact of virtual reality on proprioceptive ability by using the self-turn paradigm. The results showed that children aged 4 to 8 years made more proprioceptive errors than adults, while children aged 9 to 14 years developed proprioceptive ability to the same level as adults. In addition, it revealed that the proprioceptive error depending on visual information increases under the VR environment. This study is significant in that it provides new evidence for the debate about when proprioceptive ability develops to the same level as adults.

In this study, Bayesian statistical models were used to explore the most appropriate model. The method of statistical analysis is described in detail and very appropriate. Thus, I would appreciate that the reliability of the results is very high. However, I would like to point out that there are some problems with the experimental design.

Line 369:

The experimental task was designed that the experimenter manually rotated the chair at 90 or 180 degrees at first, but as reported in the supplement data, the actual rotation varied between trials; it distributed from 60 to 180 degrees in the case of 90 degrees, and from 100 to 270 degrees in the case of 180 degrees. Thus, there is a serious concern that the variability of chair rotation was not same between conditions or age groups. The authors should examine this possibility.

In addition, not only in the supplementary data, the information of actural chair rotation should be described in the main document. They should report the mean, variance and range of actual rotation angles for each 90 and 180 degrees.

Line 317

The authors written that vestibular information was always available while proprioception was not during passive rotation. I'd like you to know the experimental situation of passive rotation in more detail, such as whether it was done in the dark room or the participants were presented some visual information.

Line 348

The authors regard that proprioceptive information is not available in the visual condition, in which they hided visual landmarks but presented the local visual information only. However, I suppose that the proprioceptive information from the locomotor system is always available when we actively rotate the chair. Please explain more light on the logic why it becomes visual only condition without any proprioceptive information when there is no visual landmarks.

Reviewer #2: This is a very neatly designed study investigating how proprioceptive accuracy on a self turn task can be augmented by immersive virtual reality. Here, the authors found that younger children were less accurate than older children and adults on this task and made more proprioceptive errors. Additionally, proprioceptive errors increased when vision was not available, thus the authors suggest that proprioceptive is very reliant on visual information. It is heartening to see that the authors also address the limitations of their study; these do not detract from the importance of not only the findings of the current experiment, but investigations more broadly in the field of research.

As I am not an expert in Bayesian analyses, it is tricky for me to comment in detail on the analysis method used in this paper. However, it is explained clearly and logically and appears sound.

Major comments:

1. Please report the results from the ICCs when mentioning them in the video coding section (they are initially mentioned and then not described until very much later on). It makes sense to move these stats to when they are mentioned in the coding section.

2. The authors mention that due to the chair being turned manually by the experimenter, there is some variability in the end position of the chair. Is there a way to control for this in the analyses? How much did this vary between trials/participant/researchers? Make clear how many researchers acted as this experimenter (if one, there should not be a huge amount of variability across participants as the same researcher is conducting this aspect of the experiment, but if two or more researchers were completing this aspect of the study, this would potentially introduce more variability in the experiment). Also, make clear if this variability is not a big deal; perhaps restate what you are measuring (and how you are doing this) very briefly after explaining if/why the variability does not matter so much

3. The authors mention their data is non-normally distributed and positively skewed, please could they also report skewness values and normality test results (in addition to normality plots) in the SM. Was there a rationale for not normalising this dataset? Please state clearly.

4. Split Table 3 in SM by age group (i.e. number of observations by age group)

5. How was the amplitude score standardised? Z scored? Please state clearly

6. How did you come to the results on pg 18 before the model comparison section? Please state clearly the tests used, write out in full APA style, with Bonferroni corrections (if used).

Minor comments:

Pg 4, lines 63-63: “in a broader age ranges” should be “in a broader range of ages”

Pg 4, line 66: “…size, shape, relative location and dynamic.” – word missing?

Pg 9, line 208-9 “across the human developmental trajectory” – this phrasing is a little strange

Throughout: be consistent with using ‘a’ or ‘an’ before ‘HMD/head mounted device’.

Participants section: put demographic information in a table, with age range, M and SD and gender split

Throughout: consider using gender neutral pronouns e.g., “they” rather than “he/she”

Throughout: avoid using bullet points in the text. Consider using numbers or put in a table etc

Pg 18, line 468: ‘marginalized over the variable…’ – I’m not sure I understand what this means?

Table 3, 4, 5: indicate which effects are significant with *

Reviewer #3: MAJOR POINTS

• In general the ms is very long, and would benefit from some editing. E.g. way too much detail on ordering conditions.

• In the intro, I appreciate your attempt to carefully compare the senses and have no problem with you defining proprioception as the perception of the body posture, rather than as information which comes through particular sensory channels (from muscles & joints). But you are not consistent about it. Thus we have “proprioception belongs to the somatosensory system” (channel-specific); followed by discussion of visuo-proprioceptive info (focussed on the object of perception); and then back to “proprio is combined with info from the vestibular system.. and the visual system (channel-specific)”. The whole hypotheses section also uses the terms “vision” and “proprioception” in a “channel” way. Please just be consistent. And I think introducing some terminology about ‘sensory channels’ or ‘sense organs’ might help.

• Have you done a power analysis? Relatedly – did you have enough trials per condition to find effects in the inevitably noisy children’s data?

• By the end of the section ‘experimental task’ I have understood that you moved them round and had them re-find that position – but I have not understood the various sensory conditions eg IVR on or off, and the point of the markings on the room walls. Put the design/ conditions bit earlier, and clearly state the design. The vision condition is poorly described. Why is it called vision is it seems to be all about removing access to vision? I think you mean that features weren’t visible but optic flow was available through bright, nonspecific markings??

MINOR POINTS

• Vestibular info is key to the study but barely mentioned in the intro.

• Can you spell out more clearly how you arrive at your predictions from previous results (eg according to ref 30, IVR is more disruptive for kids, not adults?).

• I haven’t come across WAIC weights before – can you explain a little more?

• The analyses seem sound but I am not in a position to make really detailed judgment on them.

• You should probably cite literature on balance development e.g. Woollacott, Assainte & more recent ones.

Reviewer #4: This manuscript reports an interesting and innovative work on virtual reality and the development of multisensory integration. This study is a first step towards the understanding of these integration processes and their interactions with virtual reality across the life span. I believe that this topic is of interest to a broad audience of scientists and the general public.

I truly enjoyed the reading and, not being an expert in Bayesian analysis, I have highly appreciated the step-by-step description of the method - however I won’t be able to comment on the appropriateness of this section.

Overall, I find the manuscript logic and well structured, however there is room for improvement in clarity in a few places. Here below my suggestions for improving the manuscript:

Introduction

1. The first para of the intro is not entirely clear. Double check for grammar and writing style.

2. Unclear/inaccurate what “Synchronous multisensory stimulation creates proprioception” means. Consider an alternative term to ‘creates’ (p.3).

3. Define IVE, IVR and IVR environment. Differences and similarities among these terms are not immediate. Make sure the terms are used appropriately throughout.

4. Use abbreviations consistently, e.g. throughout the text both IVR and immersive virtual reality are used.

5. While I appreciate the practical implications of the developmental IVR works (p.6), I am unsure whether the introduction is a good place for laying them down. Consider to integrate these in the discussion instead.

6. Consider editing the first part of the sentence “Without going into philosophical reasons, ..” (l.188, p.8).

7. Consider replacing ‘ingredients’ with ‘components’ (l.194, p.8).

Methods

1. l.236, p.8; the text refers to section 2.2, but there doesn’t appear to be a numbering format in the manuscript.

2. The justification of the adult sample is slightly controversial. The age range in this study is 18-45 years. The authors add that older adults are excluded because evidences suggest deterioration of proprioceptive accuracy from age 40. Why then the sample includes adults up to 45 and not up to 40?

3. State how the sample size was determined.

4. Make use of tables rather than bullet points for description of participants’ characteristics (p.10) and for conditions (p.15).

5. Refer to Experimenter 1 and Experimenter 2 in the description of the procedure. I believe it will make the section clearer.

6. The meaning of the sentence ‘Proprioception has to be used only during recall phase, emerging form the other sensory information’ (l. 321-322, p13) is unclear.

7. At the beginning of the section ‘Conditions’ it seems like the word ‘blocks’ (l. 349 and l.351) is used to indicate two different things.

Results

1. In the Descriptives, state whether a third coder’s view was ever used, and if so in which percentage.

2. As anticipated, I’m not an expert in Bayesian analysis, however, can the authors confirm that the sample size or the number of observations are appropriate for performing 7 different models?

Finally, I’d recommend to check for typos throughout, e.g. l.510, p.21 (and), l.549, p.22 (delete ‘but’), l.611, p.25.

6. PLOS authors have the option to publish the peer review history of their article (what does this mean?). If published, this will include your full peer review and any attached files.

Reviewer #1: No

Reviewer #2: No

Reviewer #3: No

Reviewer #4: No

---

## [Author Response · Author response to Decision Letter 0]

6 Dec 2019

4th December 2019

Response to Reviewers

R: (Reviewer)

A: (Answer)

R: Reviewer #1: This study investigated the developmental process of proprioceptive ability and the impact of virtual reality on proprioceptive ability by using the self-turn paradigm. The results showed that children aged 4 to 8 years made more proprioceptive errors than adults, while children aged 9 to 14 years developed proprioceptive ability to the same level as adults. In addition, it revealed that the proprioceptive error depending on visual information increases under the VR environment. This study is significant in that it provides new evidence for the debate about when proprioceptive ability develops to the same level as adults.

A: We thank the reviewer for this accurate summary of our manuscript.

R: In this study, Bayesian statistical models were used to explore the most appropriate model. The method of statistical analysis is described in detail and very appropriate. Thus, I would appreciate that the reliability of the results is very high. However, I would like to point out that there are some problems with the experimental design.

Line 369:

The experimental task was designed that the experimenter manually rotated the chair at 90 or 180 degrees at first, but as reported in the supplement data, the actual rotation varied between trials; it distributed from 60 to 180 degrees in the case of 90 degrees, and from 100 to 270 degrees in the case of 180 degrees. Thus, there is a serious concern that the variability of chair rotation was not same between conditions or age groups. The authors should examine this possibility.

In addition, not only in the supplementary data, the information of actural chair rotation should be described in the main document. They should report the mean, variance and range of actual rotation angles for each 90 and 180 degrees.

A: Thank you for your helpful review and for this comment. It is correct that the actual rotation amplitude is an important factor in this experiment. Given the variability in actual rotation amplitude, this variable was analysed in a continuous manner, not as a dichotomous categorical variable. While the majority of rotations were relatively accurate to the aimed 90 or 180 degrees, allowing for a good estimation of accuracy in “small” and “large” rotations across conditions and groups, there were some rotations that fell outside this range. Figure 3 in the main manuscript displays the distribution of the actual amplitude in the passive rotation. In the revision process, we have also added a figure (S2 Fig.) in the Supplemental Materials document which shows the distribution of the actual self-turn in the different experimental conditions according to age group. It is possible to observe that distributions are slightly different, but all of them cover approximately the same range of values. Considering amplitude as a continuous variable and not as a dichotomous categorical variable gives us confidence that our results were not influenced by the variability in the actual rotation between experimental conditions and age groups. This decision is described in the manuscript at page 12, Lines 292-293. We prefer not to include S2 Fig. in the manuscript due to considerations of length and brevity. Indeed, as the variability of chair rotation was homogeneous across conditions and age groups, that table would not add a significant amount of information to the global distribution of the actual amplitude (Figure 3).

R: Line 317

The authors written that vestibular information was always available while proprioception was not during passive rotation. I'd like you to know the experimental situation of passive rotation in more detail, such as whether it was done in the dark room or the participants were presented some visual information.

A: Thank you for touching on this important point. We have added more detail to the Experimental Task section of the manuscript (see Lines 24-259) to describe the characteristics of the passive rotation in more detail. Ultimately, the presentation of visual information and darkness of the room varied across experimental conditions, but these factors were consistent for both the passive and active rotation within conditions. For example, in the R_P condition, the room was completely dark and thus no visual information was available during both the passive rotation and the active rotation.

We have clarified the whole section; see below for the revised version:

“We adopted a self-turn paradigm in which the experimenter rotates the chair a certain degree (passive rotation) from a start position to an end position. After each passive rotation, participants were asked to rotate back to the start position (active rotation). The position at which the participant stopped their active rotation is recorded as the return position. During the passive rotation, participants sat still and kept their feet on a footrest which rotated with the chair. To perform the active rotations, participants could use their feet on the still platform under the chair to move themselves. Within a given experimental condition, during both the encoding (passive rotation) and the recall (active rotation) phase, all sensory information were consistent. During the recall phase, proprioception derived from the active movement was involved in performing the active rotation and recalling the start position. This constitutes the accuracy measure in our task, in line with the extant literature (43-45). We did not manipulate vestibular information, which was consistent across all experimental conditions. On the other hand, we manipulated vision across the three experimental conditions as described in the following section.”

R: Line 348

The authors regard that proprioceptive information is not available in the visual condition, in which they hided visual landmarks but presented the local visual information only. However, I suppose that the proprioceptive information from the locomotor system is always available when we actively rotate the chair. Please explain more light on the logic why it becomes visual only condition without any proprioceptive information when there is no visual landmarks.

A: Many thanks for this thoughtful comment. We describe the condition as “vision only” because the visual information that was provided was proprioceptively uninformative to perform the active rotation back to the starting point. We have now expanded on the logic of this condition in the Conditions section of the manuscript as follows (Lines 271-280):

“One visual condition limited the access to proprioceptively informative visual landmarks (hiding the participants' body and the room corners) in order to disrupt proprioception, while providing a proprioceptively uninformative visual texture (a pattern of small bright clouds on the walls) (V). Indeed, after being disorientated by a passive rotation in a real environment, people could still detect the position of global landmarks (the room’s corners), while making huge errors locating surrounding objects [53]. Our intention was to disrupt proprioception through altering the visual information available, without making changes to the proprioceptive information arising from participants' body during the passive and active movements, which are consistent within participants”

The study referenced here (Wang & Spelke, 2000) indicates that global visual landmarks such as the corners of a room may be considered to be informative visual landmarks that contribute to a person’s ability to locate themselves in space, while other surrounding objects were much less accurately located following disorientation (in Wang & Spelke, these surrounding objects included a television and a pile of fabric; in our experiment, the continuous pattern of clouds on the wall which were visible in the “only vision” condition can be considered as surrounding objects). As such, our condition does not purport that proprioceptive information is “not available”, but rather that it is disrupted through the process of obscuring visual landmarks (the room’s corners, the participant’s body).

With respect to locomotion, the action through which the body as a whole moves through space, it would be expected that if locomotor information were available during the active rotation of the chair, this would be consistent across conditions given that an active rotation of the chair was made in every condition. As we now state in the manuscript, our intention was to disrupt proprioception through altering the visual information available, without making changes to locomotor information between conditions. We believe that this would have been the case, given that while conditions varied in the amount of visual and proprioceptive information that could be reliably used, they were consistent in providing a passive rotation away from the start point performed by the experimenter and an active turn back to the estimated start point performed by the participant. 

R: Reviewer #2: This is a very neatly designed study investigating how proprioceptive accuracy on a self turn task can be augmented by immersive virtual reality. Here, the authors found that younger children were less accurate than older children and adults on this task and made more proprioceptive errors. Additionally, proprioceptive errors increased when vision was not available, thus the authors suggest that proprioceptive is very reliant on visual information. It is heartening to see that the authors also address the limitations of their study; these do not detract from the importance of not only the findings of the current experiment, but investigations more broadly in the field of research.

A: We thank the reviewer for this accurate summary of our manuscript.

R: As I am not an expert in Bayesian analyses, it is tricky for me to comment in detail on the analysis method used in this paper. However, it is explained clearly and logically and appears sound.

Major comments:

1. Please report the results from the ICCs when mentioning them in the video coding section (they are initially mentioned and then not described until very much later on). It makes sense to move these stats to when they are mentioned in the coding section.

A: Thank you very much for your thoughtful review and for this sensible comment. You are absolutely correct and we have now reported the ICCs in the coding section (“Measures of task performance”) where they are initially mentioned (see Lines 330-337).

R: 2. The authors mention that due to the chair being turned manually by the experimenter, there is some variability in the end position of the chair. Is there a way to control for this in the analyses? How much did this vary between trials/participant/researchers? Make clear how many researchers acted as this experimenter (if one, there should not be a huge amount of variability across participants as the same researcher is conducting this aspect of the experiment, but if two or more researchers were completing this aspect of the study, this would potentially introduce more variability in the experiment). Also, make clear if this variability is not a big deal; perhaps restate what you are measuring (and how you are doing this) very briefly after explaining if/why the variability does not matter so much

A: Many thanks for this important comment. Each rotation began at the previous end position, so there was variability in the end position of the chair due to the variability in rotation amplitude. However, as seen in S2 Fig. (now added to the Supplemental Materials), the amplitude of turns was relatively consistent across conditions and groups, generally falling around 90 degrees for the planned smaller rotations and around 180 degrees for the planned larger rotations. We included these two approximate rotation distances in order to control for a possible learning effect (e.g. if participants were continually required to perform turns of exactly 90 degrees, they may simply become adept at reproducing this angle regardless of the experimental manipulation). Moreover, amplitude was analysed as a continuous variable so we could see how the amplitude could have affected performance. The way that amplitude may affect performance was not a main hypothesis of this experiment, although we agree that stricter control of amplitude could be a useful addition to future studies in this field which are more concerned with this variable (see also response to Reviewer 1).

We have now clarified in the manuscript that two experimenters performed the experiment at any given time, and used the labels “Experimenter 1” and “Experimenter 2” to clarify their roles. Overall, five experimenters were involved in the running of this experiment. As mentioned in the manuscript, all experimenters were trained to keep a continuous velocity in performing the angle and rotation. However, it is true that there were potentially differences in performance between experimenters, but as indicated in S2 Fig., Supplemental Materials, this variability did not differ widely across groups and conditions. Furthermore, it is important to note that this possibility was another factor in our decision to analyse amplitude as a continuous variable.

R: 3. The authors mention their data is non-normally distributed and positively skewed, please could they also report skewness values and normality test results (in addition to normality plots) in the SM. Was there a rationale for not normalising this dataset? Please state clearly.

A: We thank the reviewer for raising this issue as it allows us to clarify and discuss an important strength of the statistical approach adopted in the analysis. We decided to use Generalized Linear Models (GLMs) instead of transforming the data to properly model the characteristics of our dependent variable. GLMs allow us to model the dependent variable, specifying an appropriate probability distribution that reflects the characteristics of the data, rather than transforming the data to meet statistical assumptions (Fox, 2016; Lo & Andrews 2015; Ng & Cribbie 2017). Data transformation does not guarantee a simultaneous correction for both skewness and heteroscedasticity, whereas GLMs allow us to model non-normally distributed data by using more appropriate distributions. This results in a better fit to the data and, in turn, provides more reliable results. To clarify this point (why GLMs were used instead of transforming the data), we added the following lines to the Statistical Approach section:

(Lines: 350–360)

“Thus, participants were treated as random effects, with random intercepts that account for interpersonal variability, while the other variables are considered as fixed effects.”

“Generalized mixed-effects models were used considering the Gamma distribution, with logarithmic link function, as the probability distribution of the dependent variable. Generalized mixed-effects models allow to model non-normally distributed data using appropriate probability distributions that reflect the characteristics of the data [49]. Selecting an appropriate probability distribution provides better fit to the data and more reliable results[50].”

“Gamma distribution is advised in the case of positively skewed, non-negative data, when the variances are expected to be proportional to the square of the means [51].”

With respect to the suggestion of reporting skewness values and normality test results in the SM, we agree that the skewness value is a useful point of information to quantify the asymmetry of the data distribution. Therefore, we have added the skewness value in the SM on page 8 when observed data were presented. However, with respect to the normality test, we prefer to stress the theoretical and methodological reasons underpinning why we considered the dependent variable as non-normally distributed. The dependent variable (i.e., rotation error) was defined as the absolute difference between the start position and the return position in the self-turning task, thus, only positive values are possible. This consideration per se is sufficient to exclude the normal distribution from the possible probability distributions to represent the data. In fact, normal distribution support includes all real numbers, but in our case negative values are impossible. To correctly describe the data, we need a distribution with only positive support; in our case the Gamma distribution. In this case, adding normality test results is not necessary. Instead, we prefer to stress the importance of selecting an appropriate distribution on the basis of theoretical and methodological considerations. As reported above, we explain this decision within the “Statistical approach” section of the manuscript (Lines: 350–360).

R: 4. Split Table 3 in SM by age group (i.e. number of observations by age group)

A: Thank you for this comment. S3 Table in the Supplemental Materials contains the number of observations by age group (adult, middle, young), as suggested by the reviewer.

R: 5. How was the amplitude score standardised? Z scored? Please state clearly

A: We thank the reviewer for reporting this unclear passage in the text. Amplitude scores were standardized by subtracting the mean value from the raw scores and dividing for the standard deviation. Thus, the reviewer is correct that we obtained Z scores. We have added this information in the article to make it clear.

(Lines 397-398)

“To obtain interpretable results in the analyses, the Amplitude variable was standardized (i.e., Z scores were obtained)”

We would like to clarify that Amplitude was standardized to optimize model computation and to improve interpretability of the results. Standardizing a variable does not change the shape of the original distribution of data.

R: 6. How did you come to the results on pg 18 before the model comparison section? Please state clearly the tests used, write out in full APA style, with Bonferroni corrections (if used).

A: We thank the reviewer for reporting this mistake. In the text, the reported values are the descriptive statistics of the observed data, but we wrongly presented them as “… the marginal effect of…”. This leads the reader to think that they are the results of some tests, but actually we are only presenting descriptive statistics of the observed data according to the different variables. To avoid this misunderstanding we rephrased the paragraph as follows: 

(Lines 409-417):

“Considering the observed values according to Age, adults (M = 12.8, SD = 4.4) made less self-turn errors than older children (M = 16.4, SD = 7.5) and young children (M = 25.3, SD = 7.7). Looking at the Environment conditions, participants made less errors and were thusly more accurate in the reality condition (M = 13.9, SD = 8.0) than in the IVR condition (M = 20.2, SD=10.3). Finally, considering the different levels of the variable Perception, participants made less self-turn errors when they could rely on both vision and proprioception (M = 13.9, SD= 11.3) than when they could use only vision (M = 14.5, SD= 9.3) or proprioception (M = 22.8, SD= 14.1).”

R: Minor comments:

Pg 4, lines 63-63: “in a broader age ranges” should be “in a broader range of ages”

A: Thank you for pointing out this error. This has now been corrected in the manuscript.

R: Pg 4, line 66: “…size, shape, relative location and dynamic.” – word missing?

A: Thank you for this comment. We have removed the unclear word “dynamic”.

R: Pg 9, line 208-9 “across the human developmental trajectory” – this phrasing is a little strange

A: Thank you for pointing this out. This has been changed to read “across the lifespan”.

R: Throughout: be consistent with using ‘a’ or ‘an’ before ‘HMD/head mounted device’.

A: Thank you for this helpful comment; we have now made the use of “a head mounted device” and “an HMD” consistent throughout the manuscript. Guidance from the APA Style Blog (6th Edition) dictates that acronyms take “a” or “an” according to how they are pronounced (“a” for consonant sounds, “an” for vowel sounds), not their spelling. As such, we use “a head mounted device” because “head” starts with a consonant sound. For “HMD”, which begins with a vowel sound (“aitch em dee”), we accordingly use “an”. See the following post from the APA Style Blog, written by Jeff Hume-Pratuch and titled “Using "a" or "an" With Acronyms and Abbreviations”, for details:

https://blog.apastyle.org/apastyle/2012/04/using-a-or-an-with-acronyms-and-abbreviations.html

R: Participants section: put demographic information in a table, with age range, M and SD and gender split

A: Thank you for this useful suggestion. We have now added the demographic information in Table 1.

R: Throughout: consider using gender neutral pronouns e.g., “they” rather than “he/she”

A: Many thanks for this very helpful comment; we have now changed instances of “he/she” to “they” in the manuscript.

R: Throughout: avoid using bullet points in the text. Consider using numbers or put in a table etc

A: We appreciate this suggestion, thank you. We have now removed bullet points from the manuscript. We describe participants’ demographic features with Table 1 and the conditions with a numbered list.

R: Pg 18, line 468: ‘marginalized over the variable…’ – I’m not sure I understand what this means?

A: We apologise to the reviewer for using a misleading term. In this case “marginalisation” is not appropriate. We actually computed the descriptive statistics without taking into account the variable Amplitude. That is, to compute mean self-turn error and standard deviation according to Age, Environment, and Perception, we considered all the observations independently of the Amplitude values. We have corrected this point in the text as follows:

Lines (405-409)

“For the sake of interpretability, descriptive statistics were computed according to Age, Environment, and Perception, without taking into account the variable Amplitude (i.e., all observations in the same condition were considered independently of the Amplitude values), which will be considered later on in the analysis.” 

R: Table 3, 4, 5: indicate which effects are significant with *

A: We thank the reviewer for this note. We imagine that the reviewer suggested this to facilitate the reading of the tables and to easily identify relevant effects. However, the classical definition of a “significant” effect is rather problematic within a Bayesian framework.

In a Bayesian framework, there is no significance testing, so no p-values are computed to evaluate if effects are significant. On the contrary, the Bayesian approach evaluates which are the most plausible values of the model parameters according to the data and the prior distributions. In a Bayesian analysis, results are in the form of posterior distributions that quantify the uncertainty about the quantities of interest. From the posterior distributions, it is possible to compute the Bayesian Credible Intervals (BCIs) which represent a given portion (e.g., 95%, but it is an arbitrary choice) of the most likely values. Thus, for the sake of interpretability, an effect could be considered plausible if the value zero (or a given value of interest) is not included in this range of values. This procedure could be erroneously considered similar to the classical significance testing approach but actually its implications and interpretations are different. Among others, the Bayesian approach does not imply the dichotomous thinking about “significant” and “not significant” values typical of the Null Hypothesis Significance Testing (NHST) approach, but it allows us to think about phenomena in terms of the magnitude of evidence that supports the existence of an effect (Ortega & Navarrete, 2017). Dichotomous decision making is not meant to be the goal of Bayesian approach, where the emphasis is on the full information provided by the continuous posterior distribution (Kruschke & Liddell, 2018; Wasserstein, Schirm, & Lazar, 2019). Therefore, the distinction between presence/absence of an effect is done only to facilitate the discussion of the results, and readers should consider the full information provided by the posterior distributions represented in the graphs. To avoid the possibility that the readers would consider the results in terms of “statistically significant results”, we prefer to not report * in the tables.

Reviewer #3: MAJOR POINTS

R: In general the ms is very long, and would benefit from some editing. E.g. way too much detail on ordering conditions.

A: Thank you for providing this constructive review and for this comment. We appreciate that the original manuscript is very long and we have now removed detail on the condition order and throughout the manuscript in other places. Overall, during this review process, the manuscript has been reduced by 1.642 words and some tables have been moved to the Supplemental Materials.

R: In the intro, I appreciate your attempt to carefully compare the senses and have no problem with you defining proprioception as the perception of the body posture, rather than as information which comes through particular sensory channels (from muscles & joints). But you are not consistent about it. Thus we have “proprioception belongs to the somatosensory system” (channel-specific); followed by discussion of visuo-proprioceptive info (focussed on the object of perception); and then back to “proprio is combined with info from the vestibular system.. and the visual system (channel-specific)”. The whole hypotheses section also uses the terms “vision” and “proprioception” in a “channel” way. Please just be consistent. And I think introducing some terminology about ‘sensory channels’ or ‘sense organs’ might help.

A: Thank you for pointing to the possible confusion with this discussion of proprioception. Notably, the definition of proprioception is hugely debated in the extant literature, with different theories, authors, and papers often referring to different aspects and conceptualisations of it. Our idea here was that, just like any other sense, proprioception is influenced by the information coming from other sensory channels in multisensory integration. We do indeed describe proprioception as perception of body posture and movement, which results in a representation of the body in space. To avoid any confusion, we have now clarified that this perception/representation is formed by the information sent via body-based somatosensory proprioceptors – muscles and joints. However, we clearly state that the focus is on multisensory processes, exploring how proprioception is affected by the visual environment when vision is available. We believe that this should explain our perspective on proprioception as a distinct sensory channel when we talk about proprioceptive information, whereas the resulting perception can take different forms of complex body awareness depending on the reliability of sensory cues involved. Therefore, in the hypotheses section, proprioception is discussed as a specific sensory channel which can, for example, be coupled with vision (in our VP condition) or function independently (in our P condition).

To describe the role of different visual cues on calibrating proprioception, we introduced the term “proprioceptively informative/uninformative”. See this excerpt from the manuscript (Lines 28-38):

“While humans rely on somatosensory information to achieve proprioception in blind conditions, vision can lead to proprioception when proprioceptively informative cues are provided. Indeed, specific visual cues can be considered to be proprioceptively informative to the extent that they aid proprioception. For example, research concerning mirror therapy for phantom limb pain indicates that visual representations of the body (e.g. the lost limb) can be manipulated to induce proprioceptive sensations and perception of movement, touch, and body ownership, even with a complete absence of somatosensory input [7]. Moreover, self-motion studies show that global visual landmarks such as the corners of a room appear to be useful for proprioception, while local visual cues such as surrounding objects [7] or homogeneous visual textures and patterns [8] are not.”

R: Have you done a power analysis? Relatedly – did you have enough trials per condition to find effects in the inevitably noisy children’s data?

A: Many thanks for putting forward this question. However, the present study did not aim to evaluate specific hypotheses but was intended to explore possible relations. As such, we did not complete a power analysis before undertaking this experiment for several reasons. First and foremost, due to the small number of experiments previously conducted in this area, we did not have a good sense of the effect size we might expect. Quantifying the effect size was particularly difficult given the number of complex interactions we explored in this work. This was, first and foremost, an exploratory study in which we aimed to establish some base findings in the area of proprioceptive accuracy in an IVR- and reality-based task at different developmental stages. Our final sample included 13 younger children, 13 older children, and 23 adults. We took guidance from studies in this area in the past which have drawn informative results from smaller pools of participants. For example, in studying the ability to remember the relative location of target objects in real-world, desktop-delivered, and HMD-delivered IVR environments, Lathrop & Kaiser (2002) included eight adult participants. In a study that was very influential in the development of our own, Petrini, Caradonna, Foster, Burgess, & Nardini (2016), in which participants were required to reproduce a path they had learned in darkness, in a virtual room, or having been shown a pre-recorded version of the walk in a virtual room without moving, there were 18 adult and 15 child participants. The lack of a power analysis can be considered as a limitation of this study, but the exploratory nature of the study is declared in the abstract and stressed several times in the article to prevent readers from drawing strong conclusions. These exploratory results can now be used together with other sources of information (i.e., other studies’ results or experts’ indications) to define more accurate hypotheses and plan future confirmatory studies in this promising area of research, as suggested in the conclusions.

We included two trials per condition in order to keep the experiment sufficiently short for the younger participants, some of whom were only four years old. As our results indicate, it was possible to see differences between the age groups in this experiment.

R: By the end of the section ‘experimental task’ I have understood that you moved them round and had them re-find that position – but I have not understood the various sensory conditions eg IVR on or off, and the point of the markings on the room walls. Put the design/ conditions bit earlier, and clearly state the design. The vision condition is poorly described. Why is it called vision is it seems to be all about removing access to vision? I think you mean that features weren’t visible but optic flow was available through bright, nonspecific markings??

A: Many thanks for pointing out the potential confusion in this section. We have added detail to this section to clarify what was happening in each sensory condition (see also the response to Reviewer 1). You are correct in asserting that in the “vision only” condition, global features of the room (e.g. the room’s corners, the door, the participant’s own body) were not visible, but optic flow was available thanks to the inclusion of a continuous visual pattern of many glowing clouds on the walls. This can be seen in Figure 2b. This condition is called “V” in order to distinguish it from the other two conditions, “P” and “VP”. The “VP” condition allows accurate, reliable visual and proprioceptive input. The “P” condition removes vision entirely (the participant is blind, moving in complete darkness), allowing them to rely only on proprioception. In contrast, while we limited access to some visual information in the “V” condition, this was done in order to disrupt proprioception such that the visual information which was available was the only reliable sensory input. This explanation has been developed further in the manuscript from the original version we submitted in order to make this clearer, and to clarify the use of the HMD (and subsequent IVR) in each condition.

We have endeavoured to make the “V” condition clearer by using the description “visual texture” to describe the information provided. This refers to the visual texture of the pattern of clouds on the walls which moves through the visual field as a consequence of the participant’s movement. In our case, the visual texture of surrounding clouds on the walls moves as a consequence of participant’s movement. The “V” condition is designed to comprise “only vision” as it provides only proprioceptively uninformative visual information in the form of this texture, without visual information about the position of the body, for example. We define what is proprioceptively uninformative based on the literature described in the manuscript.

MINOR POINTS

R: Vestibular info is key to the study but barely mentioned in the intro.

A: Thank you for this comment. Vestibular information was a component of this experimental task but not one that we manipulated or addressed in this study. The vestibular information available to participants did not vary across conditions, and we were not interested to manipulate or assess vestibular information in this work. As such, we have added some information to clarify the relevance of vestibular information in the Experimental Task section, as follows (Lines 256-259):

 “We did not manipulate vestibular information, which was consistent across all experimental conditions. On the other hand, we manipulated vision across the three experimental conditions as described in the following section.”

R: Can you spell out more clearly how you arrive at your predictions from previous results (eg according to ref 30, IVR is more disruptive for kids, not adults?).

A: Many thanks for this important comment; we would be happy to expand on and clarify our thought process. The reference 30 study (Adams, Narasimham, Rieser, Creem-Regehr, Stefanucci, & Bodenheimer, 2018) found that the post-exposure effects of an IVR environment lasted longer for 8 - 12 -year-old children than for 15 - 18-year-old adolescents. The task in that experiment was a throwing task, where participants had to throw an object to a target under normal conditions, then with vision manipulated so the participant’s view was offset, and then under normal conditions again to see how long it took participants to recalibrate to the real, unadjusted environment. Although they recalibrated quickly, the younger group took longer to adapt back to their baseline performance, potentially because, as suggested by the authors, their visuo-motor system is not yet fully developed. 

In the context of our experiment, we found it interesting that the Adams et al. (2018) study indicates that the mismatch between visual and proprioceptive information in the visually manipulated condition seemed to have more enduring effects on younger children. As we suggest in our paper, we think this study provides evidence that younger children (more than adolescents or adults) could show a more enduringly affected motor performance following training in IVR, given that the effects of IVR last longer for them (they are slower to recalibrate to the “real world” environment after IVR manipulation). The referenced study didn’t play a major role in formulating our hypotheses, but it did provide an important piece of evidence that different age groups may be differently affected by IVR, and that it is necessary to shed more light on how age might affect one’s interactions with IVR. 

We rephrased the manuscript as follows (lLines 97-104)

“As with adults in previous studies [27-28], children and adolescents showed the ability to recalibrate in a few minutes. However, children re-adapted to reality significantly more slowly than adolescents, demonstrating more pronounced post-exposure effects. These findings indicate that the motor performance of children, more so than adolescents, could be driven by vision and modified by IVR. As different age groups may be differently affected by IVR, it is necessary to shed light on how age might affect one's interaction with this technology.”

R: I haven’t come across WAIC weights before – can you explain a little more?

A: Our sincere thanks for the interest in this technical aspect. WAIC can be considered as the corresponding Bayesian version of the commonly used AIC (Akaike Information Criterion). To simplify, WAIC values can be interpreted as the average error made by the models in predicting new observations. Thus, models with lower WAIC values (i.e., smaller errors) are preferred to models with higher WAIC values (i.e., greater errors). WAIC values cannot be considered in absolute terms but only compared to other WAIC values of different models. As such, results are always relative to the set of models considered in the analysis. It is possible to say that a model is the best one among a set of candidate models, but it is not possible to say that that model is absolutely the best one. There could always be another better model that was not yet considered.

However, it is difficult to understand how much a model is better than another only from WAIC values. To allow readers to better interpret the results, WAIC weights are usually presented. WAIC weights sum to 1, so they are interpreted as “an estimate of the probability that the model will make the best predictions on new data, conditional on the set of models considered” (McElreath, 2016, p.199).

To compute the WAIC weights, firstly, for each model the difference between the WAIC values of the worst model (i.e., greater WAIC) and its WAIC value is computed. Then, relative likelihood of each model with respect to the worst model is computed by taking the exponential of half of the difference previously computed (i.e., exp(diff_WAIC / 2)). Finally WAIC weights are computed by dividing the relative likelihood of each model by the sum of all the relative likelihoods previously computed. A slightly different but equivalent formula is presented by McElreath (2016, p.199; difference is given by the fact that in the formula the author used the difference between WAIC model values with the lowest WAIC value and not the difference between the highest WAIC value with WAIC model values).

We thank the reviewer for the interest, but we think that it is not appropriate to include such a detailed explanation in the article. The definition of WAIC weights was already reported in the statistical approach section of the article to allow readers to interpret the results, whereas their computation with all the steps was presented in the SM. Readers looking for more detailed information should refer to the relevant literature reported in the article. The reviewer should also consider that there is an extensive Supplemental Materials section that could be used for further clarification. In case, if the reviewer and the editor think that there is the need to include some clarification in the main text we are happy to do it.

R: The analyses seem sound but I am not in a position to make really detailed judgment on them.

A: Thank you for this comment. We appreciate that the analyses will not be the focus of this article and for this reason we have included and extended the Supplemental Materials for a more specialistic analysis. Nevertheless, we hope that they can still be enough informative in the manuscript.

R: You should probably cite literature on balance development e.g. Woollacott, Assainte & more recent ones.

A: Many thanks for this insightful comment. We are familiar with the interesting literature on balance development put forward by Woollacott, Assainte, Amblard, and others. Balance was a consideration in our experiment, and one of several reasons for our choosing a seated paradigm rather than a standing rotation. With the inclusion of this seated paradigm, we expect that any potential effects of balance would be minimised. Although we assume that more research is also needed into the parallel development of vestibular and proprioceptive systems and their integration, in this study, our aim was to concentrate on the development of proprioception by considering its integration with vision.

Reviewer #4: 

R: This manuscript reports an interesting and innovative work on virtual reality and the development of multisensory integration. This study is a first step towards the understanding of these integration processes and their interactions with virtual reality across the life span. I believe that this topic is of interest to a broad audience of scientists and the general public.

I truly enjoyed the reading and, not being an expert in Bayesian analysis, I have highly appreciated the step-by-step description of the method - however I won’t be able to comment on the appropriateness of this section.

A: We deeply thank the Reviewer for the appreciation of our work

R: Overall, I find the manuscript logic and well structured, however there is room for improvement in clarity in a few places. Here below my suggestions for improving the manuscript:

Introduction

1. The first para of the intro is not entirely clear. Double check for grammar and writing style.

A: Thank you very much for your insightful review and for this helpful comment. We do see now that the original first paragraph could use some work, and we have thusly changed it to (Lines 2-14):

“From the earliest stages of life, we develop physically, psychologically, and socially through the interaction between our genes and the environment. We experience this environment via sensory information which comes from both the external world (exteroception) and the self (interoception). Exteroception describes sensory information which comes from the environment around us (e.g. sight, hearing, touch), while interoception is the perception of our body and includes “temperature, pain, itch, tickle, sensual touch, muscular and visceral sensations, vasomotor flush, hunger, thirst” and other sensations (p. 655 [1]). This information, which comes from different, complementary sensory modalities, has to be integrated so that we can interact with and learn from the environment. The multisensory integration that follows takes time to develop and emerges in a heterochronous pattern: we rely on the various sensory modalities to different degrees at different points in the human developmental trajectory, during which the sensory modalities interact in different ways [2].”

R: 2. Unclear/inaccurate what “Synchronous multisensory stimulation creates proprioception” means. Consider an alternative term to ‘creates’ (p.3).

A: Many thanks for this comment which we very much agree with. As we were reducing the length of the paper in accordance with the other reviewers’ suggestions, this topic was no longer centrally relevant, so this line has now been removed altogether.

R: 3. Define IVE, IVR and IVR environment. Differences and similarities among these terms are not immediate. Make sure the terms are used appropriately throughout.

A: Thank you very much for this important comment. We have now used only “IVR” throughout the paper and Supplemental Materials document for the sake of clarity and consistency.

R: 4. Use abbreviations consistently, e.g. throughout the text both IVR and immersive virtual reality are used.

A: Thank you for pointing this out. We have now used IVR consistently after the original use of “immersive virtual reality”.

R: 5. While I appreciate the practical implications of the developmental IVR works (p.6), I am unsure whether the introduction is a good place for laying them down. Consider to integrate these in the discussion instead.

A: Many thanks for providing this interesting comment. We have rephrased this sentence and moved it to the Discussion, where it now reads (Lines 558-560):

“Increased knowledge in this area could have meaningful implications for fields such as IVR education, rehabilitation, and therapy, shedding light on when and how IVR interventions could be effective at different developmental stages.”

R: 6. Consider editing the first part of the sentence “Without going into philosophical reasons, ..” (l.188, p.8).

A: Thank you for providing this comment. We have now removed this phrasing from the paper altogether. 

R: 7. Consider replacing ‘ingredients’ with ‘components’ (l.194, p.8).

A: Our sincere thanks for this helpful suggestion. We have now replaced “ingredients” with “components” as suggested and agree that it is a more appropriate term.

R: Methods

1. l.236, p.8; the text refers to section 2.2, but there doesn’t appear to be a numbering format in the manuscript.

A: Thank you very much for pointing this out. You are correct, and we have now accordingly removed this reference to a numbered section.

R: 2. The justification of the adult sample is slightly controversial. The age range in this study is 18-45 years. The authors add that older adults are excluded because evidences suggest deterioration of proprioceptive accuracy from age 40. Why then the sample includes adults up to 45 and not up to 40?

A: Many thanks for offering this chance for us to clarify further our choice of age range for the adult sample. As described in the Participants section of the paper, we chose our age range based on papers which reported a deterioration of proprioceptive accuracy beginning in middle age. The papers we cited offered slightly different judgements of the specific age at which proprioceptive accuracy begins to decline. Hurley, Rees, and Newham (1998) found that proprioceptive acuity began to decline from middle age, which in their sample ranged from 50 to 64 years. Wingert, Welder, and Foo (2014) found that proprioceptive error increased with age, such that middle-aged adults (in their sample ranging from 40 to 64 years) showed significantly higher errors in joint position sense (a component of proprioception) than younger adults. However, these results are not as clear as those with older adults (in their mid-sixties and older), which show clear and consistent decreases in proprioceptive accuracy (Ingemanson, Rowe, Chan, Wolbrecht, Cramer, & Reinkensmeyer, 2016; Lee, Kwon, Son, Nam, & Kim, 2013; Pai, Rymer, Chang, & Sharma, 1997). 

Given that age is a key variable in our experiment, we wanted to remove the possibility that age-related differences in proprioceptive accuracy might affect results within the adult group. Given the evidence that these differences can begin at 40 (Hurley et al., 1998) or 50 (Wingert et al., 2014), we chose to take an average age value of these two conservative studies and limit our age range to 45 years. We accept that this justification rests on a small body of literature, but we feel that it is important to control this variable while still allowing for the inclusion of a reasonable age range of adult participants.

R: 3. State how the sample size was determined.

A: Thank you for this comment. We refer here to our response to Reviewer #3: “Due to the small number of experiments previously conducted in this area, we did not have a good sense of the effect size we might expect. Quantifying the effect size was particularly difficult given the number of complex interactions we explored in this work. This was, first and foremost, an exploratory study in which we aimed to establish some base findings in the area of proprioceptive accuracy in an IVR- and reality-based task at different developmental stages. Our final sample included 13 younger children, 13 older children, and 23 adults. We took guidance from studies in this area in the past have drawn informative results from smaller pools of participants. For example, in studying the ability to remember the relative location of target objects in real-world, desktop-delivered, and HMD-delivered IVR environments, Lathrop & Kaiser (2002) included eight adult participants. In a study that was very influential in the development of our own, Petrini, Caradonna, Foster, Burgess, & Nardini (2016), in which participants were required to reproduce a path they had learned in darkness, in a virtual room, or having been shown a pre-recorded version of the walk in a virtual room without moving, there were 18 adult and 15 child participants.”

R: 4. Make use of tables rather than bullet points for description of participants’ characteristics (p.10) and for conditions (p.15).

A: Many thanks for this suggestion. As suggested, we now make use of tables rather than bullet points for description of participants’ characteristics (p.10). We have used numbers for conditions which require a detailed description which could not fit appropriately into a table (p.15).

R: 5. Refer to Experimenter 1 and Experimenter 2 in the description of the procedure. I believe it will make the section clearer.

A: Thank you for this very useful comment. We have now done this in the manuscript.

R: 6. The meaning of the sentence ‘Proprioception has to be used only during recall phase, emerging form the other sensory information’ (l. 321-322, p13) is unclear.

A: Our thanks for pointing out this unclear phrasing. This section has now been edited and our reference to this information is as follows (Lines 249-255): 

“During the passive rotation, participants sat still and kept their feet on a footrest which rotated with the chair. To perform the active rotations, participants could use their feet on the still platform under the chair to move themselves. Within a given experimental condition, during both the encoding (passive rotation) and the recall (active rotation) phase, all sensory information were consistent. During the recall phase, proprioception derived from the active movement was involved in performing the active rotation and recalling the start position.”

R: 7. At the beginning of the section ‘Conditions’ it seems like the word ‘blocks’ (l. 349 and l.351) is used to indicate two different things.

A: Thank you for this comment. We have removed the word “blocks” to avoid confusion here and clarified what is meant in this section.

R: Results

1. In the Descriptives, state whether a third coder’s view was ever used, and if so in which percentage.

A: Thank you very much for this suggestion. We have now included the percentage as follows (Lines 323-329):

“Two independent evaluators coded the videos and entered the start and return positions in the dataset. Values which were divergent for more than two degrees were a priori considered disagreement values. That was the case for 82 out of 578 observations (14.2%). A third coder examined the video records of the disagreement values to make the final decision. In case of a disagreement value, the third coder’s value was used instead of the value that differed most from the third coder’s value.”

R: 2. As anticipated, I’m not an expert in Bayesian analysis, however, can the authors confirm that the sample size or the number of observations are appropriate for performing 7 different models?

A: Many thanks for raising this issue as it allows us to clarify an important point of the statistical approach adopted in the analysis. In a model comparison approach results are dependent on the data and the set of models considered (McElreath, 2016). This may sound trivial, however, comparing different models has nothing to do with multiple testing. In a model comparison approach, we try to explain the data observed using different mathematical models that consider different variables and relations. Models are compared using information criteria that evaluate the models’ ability to predict new data penalizing for model complexity. This allows us to identify the models that better describe the underlying data generative process, avoiding overfitting. In multiple testing, to maintain the nominal level of Type-I error, the alpha value has to be corrected for the number of tests. This requires an increased sample size to maintain adequate power. On the contrary, the number of models per se does not influence the results in a model comparison approach. The results are influenced by which models are considered. 

For example, let’s suppose we are interested in four models that reflect different theoretical perspectives and we compare them. It could be that one model is notably better than the others. However, these results are conditional on the set of models considered; the selected model is not the absolute best model. New models could be proposed that actually offer better results and we may realize that the “old best model” was actually pretty bad. 

Thus, in a model comparison approach, results depend on which models were considered and not the number of models per se. This doesn’t mean that sample size plays a minor role. As always, the larger the sample size, the more accurate and reliable the results are. In a model comparison approach, small sample sizes may not allow us to differentiate between different models. Equally good models could result because there are not enough observations to evaluate differences. As explained in the response to Reviewer #3, no power analysis was conducted given the absence of specific hypotheses or previous results in the literature. This lack of previous literature on the topic can be seen as a limitation of our work, and one of the main reasons to conduct this research in an exploratory manner. We are confident that our sample was sufficient to shed this first light on which of the models of interest was the most plausible based on our results.

R: Finally, I’d recommend to check for typos throughout, e.g. l.510, p.21 (and), l.549, p.22 (delete ‘but’), l.611, p.25.

A: Thank you very much for this comment. We have indeed proofed this manuscript thoroughly for typos before this resubmission. 

A: Our sincere thanks to the reviewers and to the editor for this constructive and thoughtful review. We have made a concerted effort to revise the manuscript to address the reviewers’ comments and are glad to return a more coherent and much improved paper.

Additional notes:

In the Supplemental Materials, both figures and tables are now labelled according to PLOS norms. The Supplemental Materials we refer to within the manuscript are listed in an appropriate section of the manuscript (Lines 620-629).

In the revision process, we added the following references to the manuscript:

Fox, J. (2016). Applied regression analysis and generalized linear models (3rd ed.). Los Angeles: SAGE.

Kruschke, J. K., & Liddell, T. M. (2018). The Bayesian New Statistics: Hypothesis testing, estimation, meta-analysis, and power analysis from a Bayesian perspective. Psychonomic Bulletin & Review, 25(1), 178–206. https://doi.org/10.3758/s13423-016-1221-4

Lo, S., & Andrews, S. (2015). To transform or not to transform: Using generalized linear mixed models to analyse reaction time data. Frontiers in Psychology, 6. https://doi.org/10.3389/fpsyg.2015.01171

McElreath, R. (2016). Statistical Rethinking: A Bayesian Course with Examples in R and Stan (1st ed.). https://doi.org/10.1201/9781315372495

Ng, V. K. Y., & Cribbie, R. A. (2017). Using the Gamma Generalized Linear Model for Modeling Continuous, Skewed and Heteroscedastic Outcomes in Psychology. Current Psychology, 36(2), 225–235. https://doi.org/10.1007/s12144-015-9404-0

Ortega, A., & Navarrete, G. (2017). Bayesian Hypothesis Testing: An Alternative to Null Hypothesis Significance Testing (NHST) in Psychology and Social Sciences. In J. P. Tejedor (Ed.), Bayesian Inference. https://doi.org/10.5772/intechopen.70230

Wasserstein, R. L., Schirm, A. L., & Lazar, N. A. (2019). Moving to a world beyond “ p < 0.05”. The American Statistician, 73(sup1), 1–19. https://doi.org/10.1080/00031305.2019.1583913

---

## [Decision Letter · Decision Letter 1]

13 Jan 2020

Proprioceptive accuracy in Immersive Virtual Reality: A developmental perspective

PONE-D-19-23517R1

Dear Dr. Farroni,

We are pleased to inform you that your manuscript has been judged scientifically suitable for publication and will be formally accepted for publication once it complies with all outstanding technical requirements.

With kind regards,

Atsushi Senju

Academic Editor

PLOS ONE

Additional Editor Comments (optional):

Reviewers' comments:

Reviewer's Responses to Questions

**Comments to the Author**

1. If the authors have adequately addressed your comments raised in a previous round of review and you feel that this manuscript is now acceptable for publication, you may indicate that here to bypass the “Comments to the Author” section, enter your conflict of interest statement in the “Confidential to Editor” section, and submit your "Accept" recommendation.

Reviewer #1: All comments have been addressed

Reviewer #2: All comments have been addressed

Reviewer #3: All comments have been addressed

Reviewer #4: All comments have been addressed

2. Is the manuscript technically sound, and do the data support the conclusions?

Reviewer #1: Yes

Reviewer #2: Yes

Reviewer #3: Yes

Reviewer #4: (No Response)

3. Has the statistical analysis been performed appropriately and rigorously? 

Reviewer #1: Yes

Reviewer #2: Yes

Reviewer #3: Yes

Reviewer #4: (No Response)

4. Have the authors made all data underlying the findings in their manuscript fully available?

Reviewer #1: Yes

Reviewer #2: Yes

Reviewer #3: Yes

Reviewer #4: (No Response)

5. Is the manuscript presented in an intelligible fashion and written in standard English?

Reviewer #1: Yes

Reviewer #2: Yes

Reviewer #3: Yes

Reviewer #4: (No Response)

6. Review Comments to the Author

Reviewer #1: The authors addressed all the issues well, and the revised manuscript deserves publication in PloS One.

Reviewer #2: The authors have address all my comments from the previous review process and I feel the manuscript is ready for publication

Reviewer #3: The authors have satisfactorily addressed my comments and I am happy to recommend the paper for publication.

Reviewer #4: (No Response)

7. PLOS authors have the option to publish the peer review history of their article (what does this mean?). If published, this will include your full peer review and any attached files.

Reviewer #1: No

Reviewer #2: No

Reviewer #3: No

Reviewer #4: No

---

## [Editor Report · Acceptance letter]

23 Jan 2020

PONE-D-19-23517R1 

Proprioceptive accuracy in Immersive Virtual Reality: A developmental perspective 

Dear Dr. Farroni:

I am pleased to inform you that your manuscript has been deemed suitable for publication in PLOS ONE. Congratulations! Your manuscript is now with our production department. 

With kind regards,

on behalf of

Dr. Atsushi Senju 

Academic Editor

PLOS ONE